# ST-VLM: KINEMATIC INSTRUCTION TUNING FOR SPATIO-TEMPORAL REASONING IN DYNAMIC VIDEOS

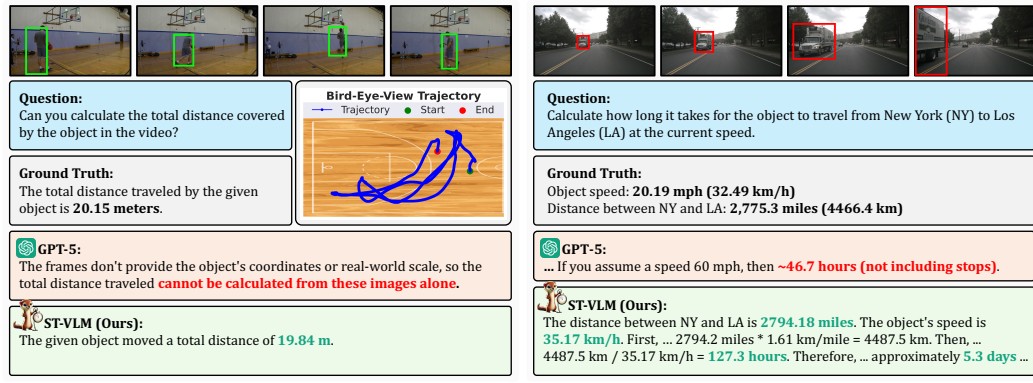

(a) A challenging case with a complex trajectory.    (b) An emergent capability of ST-VLM.

Figure 1: **Spatio-temporal reasoning in dynamic videos with moving objects.** (a) asks the model to predict the total traveled distance using *only* the video, without additional modalities such as 3D point clouds, depth map, or camera poses. The video features a basketball player moving erratically across the court, making it more difficult for the model to predict. (b) asks the model to solve multi-step reasoning questions that require integrating spatio-temporal understanding with its existing abilities (*e.g.*, commonsense knowledge, logical reasoning, arithmetic computation). Since GPT-5 lacks spatio-temporal reasoning ability, it fails to generate accurate answers. In (a), the model avoids answering in the absence of a real-world scale, whereas in (b), it assumes a speed of 60 mph, resulting in a highly inaccurate response. In contrast, ST-VLM, equipped with spatio-temporal reasoning with the proposed STKit dataset, consistently provides accurate answers in both cases.

## ABSTRACT

Spatio-temporal reasoning is essential for understanding real-world environments in various fields, *e.g.*, autonomous driving and sports analytics. While recent advances have strengthened the spatial reasoning abilities of Vision-Language Models (VLMs) through large-scale training data, these models still struggle with kinematic aspects such as traveled distance and speed of moving objects. To bridge this gap, we construct a spatio-temporal reasoning dataset and benchmark for kinematic instruction tuning, referred to as **STKit** and **STKit-Bench**. They consist of real-world videos with 3D annotations that capture object motion dynamics, including traveled distance, speed, movement direction, inter-object distance comparisons, and relative movement direction. To further scale data construction to videos without 3D annotations, we propose an automatic pipeline for generating pseudo-labels via 4D reconstruction at a real-world scale. Building on this kinematic instruction tuning data, we introduce **ST-VLM**, a VLM enhanced for spatio-temporal reasoning, which achieves strong performance on STKit-Bench. Moreover, ST-VLM generalizes robustly across diverse domains and tasks, outperforming baselines on comprehensive spatio-temporal reasoning benchmarks. Finally, by integrating learned spatio-temporal reasoning with existing abilities, ST-VLM enables complex multi-step reasoning grounded in kinematics.

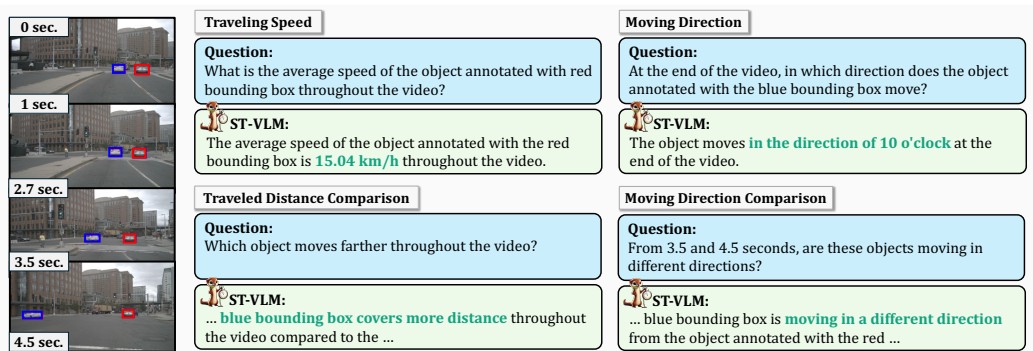

Figure 2: **Several task examples from the proposed STKit along with predictions of ST-VLM**.

# 1 INTRODUCTION

Spatio-temporal reasoning is the ability to infer how objects move and interact over time within dynamic environments from visual evidence. For example, when analyzing a video of two cars driving on the road, it involves estimating kinematic quantities such as which car moves faster, what their movement directions are, and the precise speed of a specific car. This ability is essential in a wide range of applications, including autonomous driving, sports analytics, augmented/virtual reality, and embodied AI. However, even advanced AI models still struggle to measure kinematic quantities requiring 3D/4D understanding, as shown in Fig. 1a, where GPT-5 fails to estimate a basketball player's traveled distance in a short video. Furthermore, these models often rely on language priors instead of genuinely analyzing the video's underlying kinematics. In Fig. 1b, GPT-5 simply assumes a speed of 60 mph for a car to answer the question. These observations expose a fundamental gap in the ability of existing Vision-Language Models (VLMs) to perform spatio-temporal reasoning.

Current VLMs are mostly trained on high-level vision tasks, *e.g.*, classifying object attributes or localizing 2D coordinates (Yu et al., 2016; Krishna et al., 2017). In contrast, spatio-temporal reasoning requires 3D/4D information (*e.g.*, point clouds, metric depths, and camera extrinsics). These signals are inherently low-level and difficult for VLMs to leverage effectively. To overcome this limitation, recent studies (Chen et al., 2024; Cheng et al., 2024a) have attempted to enhance spatial reasoning in image-based VLMs through large-scale datasets annotated with static geometric cues such as object sizes and locations. However, these efforts remain restricted to static scenes and cannot capture how objects evolve over time. As a result, temporal dynamics, *e.g.*, motion patterns and trajectory evolution, are left unaddressed, even though they are fundamental for kinematic understanding. This limitation motivates the need for large-scale video datasets annotated with dynamic geometric information, enabling video-based VLMs to reason over kinematics in evolving environments.

To this end, we propose ST-VLM, a VLM equipped with enhanced spatio-temporal reasoning capabilities grounded in kinematic information. To train and evaluate ST-VLM, we introduce STKit and STKit-Bench, **s**patio-**t**emporal reasoning datasets and benchmarks specifically designed for **k**inematic **i**nstruction **t**uning. These datasets comprise seven fundamental tasks that require kinematic reasoning, such as estimating traveled distance and speed (see Fig. 2 for examples). To ensure high-quality kinematic instructions, the datasets are constructed from 3D annotations, including driving videos (Wilson et al., 2023; Caesar et al., 2020) with LiDAR-based point clouds and sports videos (Grauman et al., 2024) with SLAM-based point clouds estimated from AR devices (Engel et al., 2023). Since acquiring point cloud-labeled training videos is challenging, we further develop a pseudo-labeling pipeline based on 4D reconstruction from unlabeled videos (Yu et al., 2020; Zhang et al., 2024c; Li et al., 2021). By training on both labeled and pseudo-labeled kinematic instruction data, ST-VLM enables complex reasoning that integrates spatio-temporal reasoning with its pretrained knowledge. For example, as shown in Fig. 1b, ST-VLM can answer questions that require integrating commonsense knowledge (distances between cities), kinematic estimation (speed), logical reasoning (time = distance/speed), and arithmetic computation. These emergent capabilities are seamlessly unified when spatio-temporal reasoning is incorporated into the model, even without explicit training for complex reasoning.

Table 1: **Overview of kinematic instructions.** A common prompt is prepended to each task, providing contextual information about the video: "The video lasts for $t$ seconds, and $n$ frames are uniformly sampled from it. These frames are located at $t_1, t_2 \ldots, t_n$ seconds. There are $k$ objects annotated with [COLOR] bounding boxes in the video."

| Main Categories | Subcategories | Tasks | Descriptions |
|---|---|---|---|
| Single Object | Distance | Traveled Distance | Predict the total traveled distance of the object given the timestamp. *e.g.*, Can you calculate the total distance the object traveled between [START] and [END] seconds? |
| | | Traveling Speed | Predict the average traveling speed of the object given the timestamp. *e.g.*, Tell me the object's average speed throughout the video. |
| | Direction | Movement Direction | Predict the movement direction of the object at the end of the video. *e.g.*, What direction does the object travel at the end of the video? |
| | | Direction Timestamp | Predict the timestamp when the object moves in the given direction. *e.g.*, Describe the timestamp when the object moves in the [DIRECTION] o'clock direction. |
| Multiple Objects | Distance | Traveled Distance Comparison | Compare which object has traveled farther (or less). *e.g.*, Which object travels a greater distance in the video? |
| | | Traveling Speed Comparison | Compare which object has traveled faster (or slower). *e.g.*, Which object moves faster throughout the video? |
| | Direction | Movement Direction Comparison | Compare whether objects are moving in the same direction or not. *e.g.*, Is object A moving in the same direction as object B in the video? |

In summary, our contributions are threefold:

- We introduce STKit and STKit-Bench, a new dataset and benchmark designed to endow VLMs with kinematic understanding in dynamic videos, enabling spatio-temporal reasoning over quantities such as traveled distance and movement direction.

- To address the scarcity of 3D-annotated data, we propose a pseudo-label generation pipeline that leverages 4D reconstruction from unlabeled videos.

- We present ST-VLM, which significantly surpasses GPT-5 by 25.6% on STKit-Bench with strong spatio-temporal reasoning. Our in-depth analyses demonstrate that ST-VLM excels in complex reasoning about object kinematics across various scenarios.

## 2 RELATED WORK

**Vision-Language Models (VLMs).** Recent VLMs have demonstrated strong perception and reasoning across a wide range of image (Li et al., 2024a; Xu et al., 2024; Wang et al., 2024a; Lin et al., 2024) and video (Zhang et al., 2024b; Wang et al., 2024c; Cheng et al., 2024b; Maaz et al., 2023; Li et al., 2024b) tasks, powered by LLMs. However, they struggle with 3D geometry (Liu et al., 2024). To mitigate this, spatial-aware image-based VLMs (Liu et al., 2025; Cai et al., 2025a; Yang et al., 2025; Cai et al., 2025a; Cheng et al., 2024a; Chen et al., 2024) improve spatial reasoning, such as SpatialCoT (Liu et al., 2025) with chain-of-thought (CoT) spatial grounding. Video-based VLMs (Cheng et al., 2024b; Li et al., 2025; Bhattacharyya et al., 2024) have begun to explore spatio-temporal reasoning for domains like autonomous driving (Zhou et al., 2024b; Wang et al., 2024b; Ma et al., 2024) and embodied AI (Huang et al., 2024; Cai et al., 2025a). For example, Bhattacharyya et al. (2024) propose an elegant three-step video reasoning framework (Look, Remember, Reason) incorporating a two-stream video encoder and spatio-temporal attention. In parallel, agent-based systems (Shen et al., 2023; Gupta & Kembhavi, 2023) achieve strong performance in 2D tasks (Wang & Ke, 2024; Lee et al., 2024) by chaining specialist modules whose outputs (*e.g.*, object categories, 2D bounding box coordinates) are directly interpretable by VLMs. However, VLMs remain unable to interpret low-level 3D/4D signals, limiting generalization beyond 2D domains. We address this by proposing a new video-based VLM with spatio-temporal reasoning capabilities, which directly estimates object kinematics such as traveled distance and movement direction.

**Spatio-temporal reasoning datasets.** Several datasets have been proposed in the literature (Li et al., 2025; Lei et al., 2020; Zhang et al., 2020; Zhou et al., 2025) to improve the video-based VLMs' spatio-temporal reasoning ability. ST-Align (Li et al., 2025) is a video instruction dataset that requires localizing 2D coordinates over time, whereas VidSTG (Zhang et al., 2020) focuses on spatio-temporal grounding given a query sentence. Also, several benchmarks have been introduced to evaluate video-based VLMs' spatio-temporal reasoning abilities in the general domain (Li et al., 2024b; Fu et al., 2024; Liang et al., 2025), embodied AI (Zhang et al., 2024a), and autonomous driving (Zhou et al., 2024b; Wang et al., 2024b; Sima et al., 2024). For example, Liang et al.

Figure 3: **Pseudo-label generation pipeline.** In the geometric reconstruction branch, a canonicalized 4D scene is reconstructed using MonST3R and Metric3D v2. The semantic understanding branch extracts object bounding boxes, segmentation masks, and trajectories via Grounded-SAM2. By integrating the two branches, 2D object masks are lifted into 3D, and trajectories are derived by tracking 3D centroids within the reconstructed 4D scene. Finally, a three-stage filtering strategy is applied to generate high-quality QA pairs.

(2025) introduce a novel pixel-level fine-grained spatio-temporal grounding benchmark in egocentric videos. Also, Sima et al. (2024) focus on driving-specific scenarios, such as planning and decision-making, while our dataset targets core kinematic reasoning. Concurrent with our work, VLM4D (Zhou et al., 2025) introduced a video benchmark with 4D features designed to evaluate the spatio-temporal reasoning capabilities of VLMs. However, these datasets and benchmarks do not explicitly take into account kinematics in dynamic videos, while we present instruction-tuning data annotated with kinematic information.

## 3 METHOD

We aim to infuse VLMs with spatio-temporal reasoning abilities through kinematic instruction tuning data, STKit. In Sec. 3.1, we introduce seven tasks to categorize kinematic instructions of STKit. We then present a kinematic grounding framework for generating QA pairs in STKit, using dynamic videos annotated with 3D point clouds in Sec. 3.2. To address the bottleneck of limited 3D annotations, we propose a pseudo-labeling pipeline that leverages 4D reconstruction on unlabeled videos, as detailed in Sec. 3.3. Finally, in Sec . 3.4, we train ST-VLM with STKit based on both 3D-annotated and pseudo-labeled data.

### 3.1 KINEMATIC INSTRUCTIONS

We introduce STKit, a kinematic instruction tuning dataset designed to enhance VLMs' spatio-temporal reasoning capabilities. The dataset includes instructions for measuring kinematic quantities in dynamic videos, such as object trajectories, traveled distances, and movement directions. To cover diverse kinematic aspects, we define seven tasks grouped into two categories, *Single Objects* and *Multiple Objects*, each further subdivided into *Distance* and *Direction* (see Tab. 1 for details). The tasks require the model to capture both *absolute* measures (distance and direction of an object's movement) and *relative* measures (comparisons across multiple objects). Solving them necessitates inferring spatial information (*e.g.*, object locations) and temporal information (*e.g.*, object dynamics), thereby fostering complex spatio-temporal reasoning built upon the prior knowledge of LLMs.

### 3.2 KINEMATIC GROUNDING IN DYNAMIC VIDEOS WITH 3D ANNOTATIONS

Generating QA pairs for STKit requires grounding object kinematics in dynamic videos. To this end, we consider diverse dynamic scenarios, including autonomous driving and outdoor sports (*e.g.*, football and basketball). Specifically, for driving, we use datasets such as Argoverse2 (Wilson et al., 2023) and NuScenes (Caesar et al., 2020), which provide LiDAR-based 3D object coordinates at a real-world scale for each timestamp. For sports, we incorporate Ego-Exo4D (Grauman et al., 2024), captured with wearable AR devices (Engel et al., 2023) that record both RGB images and IMU

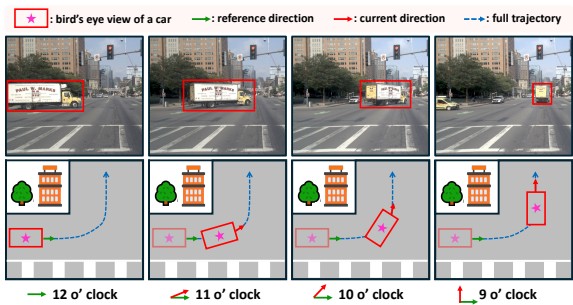

Figure 4: **Movement directions as clockwise directions.**

Table 2: **Data composition of STKit.** We extract 21K dynamic videos and generate a total of 63K kinematic instructions from six datasets, covering autonomous driving (AD), sports, and general domains. For videos without 3D annotations, we generate pseudo-labels via 4D reconstruction.

| Dataset | # QA pairs | # Videos | Domain | 3D annot. |
|---|---|---|---|---|
| NuScenes | 13K | 4K | AD | LiDAR |
| Argoverse2 | 8K | 0.6K | AD | LiDAR |
| Ego-Exo4D | 6K | 0.8K | Sports | VIO/SLAM |
| BDD100K | 35.2K | 15K | AD | pseudo-label |
| LLaVA-Video | 0.5K | 0.3K | General | pseudo-label |
| MultiSports | 0.3K | 0.2K | Sports | pseudo-label |

signals, enabling accurate 3D trajectory estimation via Visual-Inertial Odometry (VIO) and SLAM. These trajectories are used as the practical ground truth (GT).

For each annotated object in the video, we obtain its 3D centroid and 3D bounding box coordinates in world space at every timestamp. Using the 3D center coordinate $\mathbf{P}_t^{(i)}$ of the $i$-th object at time $t$, we construct trajectories by sampling centers at 0.5-second intervals over 40-frame videos, covering up to 20 seconds. The traveled distance of object $i$ between $s$ and $e$ seconds is computed as the cumulative sum of distances between consecutive frames, *i.e.*, $\sum_{t=s}^{e-1} \|\mathbf{P}_{t+1}^{(i)} - \mathbf{P}_t^{(i)}\|_2$. The average speed is also obtained by dividing the traveled distance by the duration $e - s$.

For movement direction, we first define a reference direction for each object using its initial motion, derived from the displacement between the first two frames in which it appears, *i.e.*, $\mathbf{P}_{s+1}^{(i)} - \mathbf{P}_s^{(i)}$. Subsequent movement directions are then expressed as relative angles with respect to this reference vector. However, describing directions with angles is not intuitive, as humans typically do not use exact degrees. To make this more accessible for both humans and VLMs, we discretize the calculated angles into clockwise directions. Specifically, the initial reference direction is aligned with 12 o'clock, and subsequent directions are expressed relative to this reference, as illustrated in Fig. 4. Due to the highly complex 3D trajectories in the sports domain, we exclude the movement direction category from that domain (see Fig. 1a for an example trajectory). This results in a total of 27K high-quality samples with 3D GT annotations from NuScenes, Argoverse2, and Ego-Exo4D.

### 3.3 PSEUDO-LABELING FOR UNLABELED DYNAMIC VIDEOS

To mitigate the scarcity of 3D GT annotations and extend STKit to broader domains, we propose a pseudo-labeling pipeline that leverages 4D reconstruction on unlabeled video datasets. Specifically, we use BDD100K (Yu et al., 2020) for autonomous driving, MultiSports (Li et al., 2021) for sports, and LLaVA-Video (Zhang et al., 2024c) for the general domain. Building on recent advances in *geometric reconstruction* and *semantic understanding*, we reconstruct 4D scenes by lifting segmented objects from 2D frames into 3D space. This extends the kinematic grounding described in Sec. 3.2 to unlabeled videos. An overview of the pseudo-labeling pipeline is presented in Fig. 3.

**Geometric reconstruction branch.** For 4D reconstruction on unlabeled videos, we employ MonST3R (Zhang et al., 2025b), a framework that estimates scene geometry including depth and camera intrinsics/extrinsics, even in dynamic videos with moving objects. However, the space reconstructed by MonST3R is not aligned with real-world scale, since it lacks a fixed depth reference, resulting in reconstructions that are accurate in shape but arbitrary in size. This scale misalignment poses a significant challenge for spatio-temporal reasoning tasks such as estimating traveled distances and speeds. To resolve this issue, we incorporate Metric3D v2 (Hu et al., 2024) to obtain absolute metric depth at a real-world scale. Specifically, we canonicalize the reconstructed 4D scene by rescaling MonST3R's depth estimates to match the metric depths provided by Metric3D v2.

**Semantic understanding branch.** We extract bounding boxes, segmentation masks, and trajectories of target objects using the open-vocabulary video semantic understanding model, Grounded-SAM2 (Ren et al., 2024). We focus on moving object classes of interest, including automobiles (*e.g.*, cars, buses, trucks, motorcycles, bicycles) and humans.

**Kinematic grounding in canonicalized 4D scene.** By integrating the outputs from the two branches, the 2D segmentation masks of selected objects are lifted into 3D point clouds within the canonicalized 4D reconstructed scene. We then compute each object's traveled distance, speed, and

movement direction by tracking its 3D centroid across frames, following the procedure in Sec. 3.2. Further details of the pseudo-labeling pipeline are provided in Sec. A.1.

**Filtering strategy.** Due to inherent limitations of monocular 4D reconstruction, *e.g.*, partial visibility and viewpoint constraints, we introduce a three-stage filtering strategy. *Rule-based filtering* applies predefined heuristics to discard poorly reconstructed scenes. Specifically, we remove scenes with insufficient point clouds or extremely small objects, detect outliers in 3D centroid trajectories, and apply trajectory smoothing to recover natural motion patterns. *General model-based filtering* leverages a VLM to exclude scenes with occluded objects or significant camera motion, while also assessing object detection and tracking quality to eliminate low-quality cases. *Task-specific model-based filtering* utilizes the model trained only on 3D-annotated (GT) data introduced in Sec. 3.2. This model filters out low-quality pseudo-labeled samples based on their likelihood scores.

After the three-stage filtering, the number of low-quality samples is significantly reduced. For example, scenes with occluded objects are reduced from 12,035 to 174, and those with failed object detections from 9,278 to 206. As a result, we obtain a total of 36K high-quality pseudo-labeled samples. To assess the reliability of pseudo-labels, we compare the computed trajectories against GT trajectories from the 3D-annotated dataset, NuScenes. For the traveled distance, the mean error rate decreases from 207% to 29% after applying the three-stage filtering strategy, where the error rate is defined as $\frac{|\text{Pred}-\text{GT}|}{\text{GT}} \times 100$. This demonstrates that our filtering strategy substantially improves pseudo-label quality. Further details and analysis of the filtering strategy are provided in Sec. B.

### 3.4 KINEMATIC INSTRUCTION TUNING

Based on 27K 3D sensor-annotated samples (Sec. 3.2) and 36K pseudo-labeled samples generated via 4D reconstruction (Sec. 3.3), we construct STKit for kinematic instruction tuning using predefined templates. We provide the templates for each task in Sec. F and present the detailed data composition of STKit in Tab. 2. To specify the object of interest, we overlay a bounding box on each frame as a visual prompt and provide additional textual context, including frame timestamps and bounding-box color, as input to the VLMs. We further blend STKit with subsets of general instruction tuning datasets, LLaVA-Video (Zhang et al., 2024c), LLaVA-OneVision (Li et al., 2024a), and OpenSpatialDataset (Cheng et al., 2024a), to train ST-VLM. Our model is initialized from the pretrained LLaVA-OneVision 7B. Through this integration, we empirically observe that ST-VLM exhibits emergent capabilities, combining pretrained knowledge with newly acquired spatio-temporal reasoning to support complex multi-step reasoning. Detailed analyses are presented in Sec. 6.3.

## 4 STKIT-BENCH

Since no benchmark exists for evaluating the spatio-temporal reasoning capabilities of general VLMs, particularly in object kinematics, we introduce STKit-Bench, which comprises four datasets spanning autonomous driving and sports. For autonomous driving, we use NuPlan (Caesar et al., 2021), NuScenes (Caesar et al., 2020), and Argoverse2 (Wilson et al., 2023), all of which provide LiDAR-based annotations, with NuPlan serving as an unseen dataset for robust evaluation. For sports, we adopt Ego-Exo4D (Grauman et al., 2024), which includes SLAM-based annotations. Following the official validation splits, STKit-Bench consists of 74.8% NuPlan, 12.5% NuScenes, 5.6% Argoverse2, and 7.1% Ego-Exo4D. Each task contains 200 QA pairs, resulting in a total of 1,400 QA pairs. To mitigate the long-tailed label distribution in QA pairs, we balance the number of samples across labels to ensure fair evaluation. Details of STKit-Bench are provided in Sec. E.

For evaluation, we use GPT-5-nano to extract predictions from natural language responses. We then compare each prediction $\hat{y}$ with the GT $y$ by using the following evaluation metrics:
(1) *Traveled Distance* and (2) *Traveling Speed*: Accuracy (correct if $y \times 0.75 \leq \hat{y} \leq y \times 1.25$) and MAE ($|y - \hat{y}|$).
(3) *Movement Direction*: Accuracy (correct if $y = \hat{y}$) and MAE ($\min(|y - \hat{y}|, 12 - |y - \hat{y}|)$, in clockwise directions).
(4) *Direction Timestamp*: Accuracy (correct if $\text{IoU}(y, \hat{y}) \geq 0.5$) and IoU.
(5) *Traveled Distance Comparison*, (6) *Traveling Speed Comparison*, and (7) *Movement Direction Comparison*: Accuracy (binary).

Table 3: **Results on STKit-Bench.** The average accuracy is reported in the last column.

| Models | Single Object (absolute) | | | | | | | | Multiple Objects (relative) | | | Average |
| | Traveled Distance | | Traveling Speed | | Movement Direction | | Direction Timestamp | | Travel. Distance Comparison | Travel. Speed Comparison | Move. Direction Comparison | |
| | Acc↑ | MAE↓ (m) | Acc↑ | MAE↓ (km/h) | Acc↑ | MAE↓ (clock) | Acc↑ | IoU↑ | Acc↑ | Acc↑ | Acc↑ | Acc↑ |
|---|---|---|---|---|---|---|---|---|---|---|---|---|
| *closed-source models* | | | | | | | | | | | | |
| GPT-4V | 8.0 | 48.3 | 10.0 | 33.2 | 9.0 | 2.7 | 28.0 | 0.27 | 51.0 | 49.5 | 54.0 | 29.9 |
| GPT-4o | 2.0 | 36.7 | 0.5 | 23.4 | 16.0 | 2.7 | 5.5 | 0.08 | 54.5 | 56.0 | 58.5 | 27.6 |
| GPT-5 | 1.0 | 45.3 | 5.5 | 32.5 | 12.0 | 2.23 | 8.5 | 0.11 | 65.5 | 63.0 | 73.0 | 32.6 |
| Gemini-2.5-Flash | 16.5 | 37.8 | 14.5 | 47.1 | 20.5 | 2.04 | 34.0 | 0.33 | 59.0 | 57.5 | 68.5 | 38.6 |
| Gemini-2.5-Pro | 10.5 | 38.0 | 6.5 | 37.7 | 10.0 | 2.33 | 34.0 | 0.32 | 64.0 | 62.5 | 73.5 | 37.3 |
| *open-source models* | | | | | | | | | | | | |
| VideoLLaMA3-7B (Zhang et al., 2025a) | 12.5 | 55.9 | 29.5 | 20.4 | 16.5 | 2.0 | 10.5 | 0.15 | 44.0 | 40.5 | 55.0 | 29.8 |
| Qwen2.5-VL-7B (Bai et al., 2025) | 7.0 | 60.0 | 12.0 | 84.4 | 15.5 | 2.14 | 5.0 | 0.05 | 51.0 | 45.0 | 49.5 | 26.4 |
| InternVL3-8B (Zhu et al., 2025) | 10.0 | 55.5 | 8.0 | 55.21 | 16.0 | 1.99 | 15.0 | 0.16 | 52.0 | 55.0 | 57.0 | 30.4 |
| InternVideo2.5-8B (Wang et al., 2025) | 5.5 | 367.2 | 7.5 | 31.0 | 8.5 | 3.0 | 15.0 | 0.16 | 47.5 | 49.0 | 55.5 | 26.9 |
| VideoChat-Flash-7B (Li et al., 2024c) | 8.0 | 43.79 | 14.0 | 24.8 | 10.0 | 2.9 | 23.0 | 0.25 | 51.0 | 46.0 | 47.5 | 28.5 |
| LLaVA-Video-7B (Zhang et al., 2024c) | 9.5 | 50.0 | 9.5 | 22.7 | 13.0 | 2.3 | 7.0 | 0.08 | 49.5 | 45.0 | 46.5 | 25.7 |
| LLaVA-OneVision-7B (Li et al., 2024a) | 11.5 | 54.6 | 6.0 | 25.3 | 5.0 | 2.0 | 22.5 | 0.22 | 42.5 | 52.5 | 45.0 | 26.4 |
| ST-VLM-7B (**Ours**) | **42.0** | **20.4** | **44.5** | **11.7** | **31.0** | **1.7** | **67.5** | **0.56** | **74.0** | **74.5** | **74.0** | **58.2** |

# 5 EXPERIMENTS

We compare ST-VLM with baselines on STKit-Bench under various settings for robust evaluation.

## 5.1 EXPERIMENTAL SETTINGS

**Baselines.** We evaluate closed-source proprietary models, including GPT-4V, GPT-4o, GPT-5, Gemini-2.5-Flash, and Gemini-2.5-Pro. In addition, we consider a range of open-source video-based VLMs: VideoLLaMA3 (Zhang et al., 2025a), Qwen2.5-VL (Bai et al., 2025), InternVL3 (Zhu et al., 2025), InternVideo2.5 (Wang et al., 2025), VideoChat-Flash (Li et al., 2024c), LLaVA-Video (Zhang et al., 2024c), and LLaVA-OneVision (Li et al., 2024a).

**Implementation details.** For instruction tuning, we construct the training set by blending STKit (63K) with subsets of 500K samples from LLaVA-Video, 500K samples from LLaVA-OneVision, and 100K samples from OpenSpatialDataset. 4D scene reconstruction using MonST3R takes approximately 400 seconds per video on a single A6000 GPU. We train our model for one epoch with a batch size of 128, adopting a cosine learning rate scheduler with an initial learning rate of 1e-5. The full training requires two weeks on 8 × A6000 GPUs. Further details are provided in Sec. A.2.

## 5.2 QUANTITATIVE RESULTS

Tab. 3 reports results on STKit-Bench, comparing ST-VLM against baseline VLMs. Proprietary models, including the GPT and Gemini series, exhibit weak spatio-temporal reasoning on this benchmark. For example, in Traveled Distance, Gemini-2.5-Pro attains only 16.5% accuracy with a mean absolute error (MAE) of 37.8, corresponding to an average deviation of 37.8 m from the GT distance. Open-source models also face challenges; for instance, LLaVA-OneVision, the initialization for ST-VLM, achieves only 26.4% average accuracy. In contrast, ST-VLM surpasses all baselines across the seven tasks by a substantial margin, achieving a 31.8% higher average accuracy than LLaVA-OneVision. Specifically, ST-VLM attains 44.5% accuracy on Traveling Speed with an average deviation of 11.7 km/h, showing kinematic reasoning capabilities absent in previous models.

## 5.3 ROBUST EVALUATION ACROSS DIVERSE SETTINGS

To ensure a fair and rigorous evaluation, we consider two experimental settings. First, since ST-VLM implicitly learns 3D geometric priors from 3D ground-truth annotations and pseudo-labels from 4D reconstruction, we compare it against a baseline augmented with the same information. Specifically, we evaluate GPT-5 in a few-shot setting where specialized modules (*i.e.*, MonST3R and Metric3D v2) provide camera poses as textual prompts and depth maps as auxiliary image inputs. As shown in Tab. 4, incorporating few-shot examples and geometric priors yields no performance gain, still leaving GPT-5 far behind our ST-VLM. Even with additional context, we observe that GPT-5 tends to replicate the GT values from in-context examples rather than engaging in genuine reasoning (see Sec. D.5 for the case study). This suggests that even advanced AI agents remain limited in handling 3D representations, highlighting the need for instruction data tailored to kinematic reasoning. Second, we test the robustness of ST-VLM to question variations on STKit-Bench by paraphrasing

Table 4: **Comparison with GPT-5.** We provide $N$ few-shot examples and additional geometric contexts, *i.e.*, camera extrinsics and depth maps, to GPT-5 and report the average accuracy.

| Models | GPT-5 | | | | | | ST-VLM |
|---|---|---|---|---|---|---|---|
| $N$-shots | 0 | 1 | 3 | 0 | 1 | 3 | - |
| geometric | ✗ | ✗ | ✗ | ✓ | ✓ | ✓ | - |
| avg. acc. | 32.6 | 25.8 | 22.9 | 32.9 | 24.6 | 32.4 | **58.2** |

Table 5: **Results on paraphrased STKit-Bench.** We report the average accuracy on both the original and paraphrased questions.

| Models | original | paraphrased |
|---|---|---|
| GPT-5 | 32.6 | 32.2 |
| LLaVA-OneVision | 26.4 | 28.6 |
| ST-VLM | **58.2** | **57.1** |

Table 6: **Ablation studies on pseudo-labeled data and the filtering strategy.**

| GT label | Pseudo-label | Filtering | avg. acc. |
|---|---|---|---|
| - | - | - | 26.4 |
| ✓ | - | - | 52.6 |
| - | ✓ | - | 40.4 |
| - | ✓ | ✓ | 46.1 |
| ✓ | ✓ | ✓ | 59.6 |

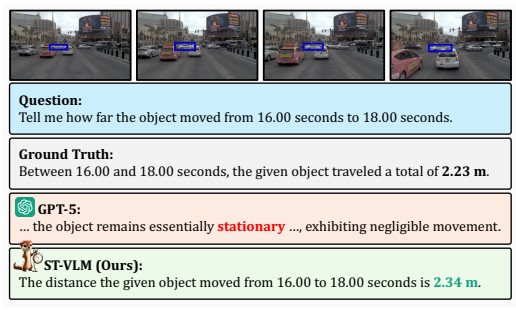

(a) Multi- and small-object scenario (22 objects).  (b) Object occlusion scenario (2nd frame).

Figure 5: **Qualitative results on STKit-Bench.**

questions using GPT-5. As shown in Tab. 5, ST-VLM consistently maintains strong performance across paraphrased questions, outperforming all baselines by a clear margin.

### 5.4 QUALITATIVE RESULTS

Fig. 5 presents qualitative results on STKit-Bench to show the robustness of ST-VLM across diverse scenarios. In Fig. 5a, ST-VLM accurately predicts the traveled distance even in videos containing numerous small objects (22 in total). Moreover, as shown in Fig. 5b, ST-VLM provides correct predictions despite partial temporal occlusions, underscoring its robustness in challenging real-world settings. A detailed quantitative analysis across various scenarios is presented in Sec. C.1.

## 6 ANALYSIS

In this section, we provide in-depth analyses to answer the following research questions:
**Q1.** How effective are pseudo-labels and the filtering strategy?
**Q2.** Does the spatio-temporal reasoning of ST-VLM generalize across various domains and tasks?
**Q3.** Does ST-VLM exhibit emergent capabilities, combining spatio-temporal reasoning (*learned* ability) with LLM's knowledge (*existing* ability) within multi-step reasoning?

### 6.1 ANALYSIS ON PSEUDO-LABELS

Tab. 6 presents ablation studies on pseudo-labeled data and the filtering strategy to discuss **Q1**. Training only with GT-labeled data substantially improves performance from 26.4% to 52.6%. Also, training only with pseudo-labeled data improves performance by 14.0%, while the filtering strategy provides an additional 5.7% gain. Finally, incorporating both GT-labeled and pseudo-labeled data along with our filtering strategy shows a remarkable performance gain, underscoring the effectiveness of our 4D reconstruction-based pseudo-labeling and filtering pipeline. A detailed analysis of the pseudo-labels and the filtering strategy is presented in Sec. B.

### 6.2 GENERALIZED SPATIO-TEMPORAL UNDERSTANDING

We assess the generalization ability of ST-VLM's spatio-temporal reasoning on comprehensive video benchmarks to answer **Q2**. As shown in Tab. 7, ST-VLM trained with STKit outperforms

Table 7: **Results on comprehensive video benchmarks.**

| Models | PerceptionTest val | MVBench test | VideoMME w/o & w/ subtitle | MLVU m-avg | NExT-QA test | Avg. acc |
|---|---|---|---|---|---|---|
| GPT-4o | - | - | 71.9 & 77.2 | 64.6 | - | 71.2 |
| Gemini-1.5-Pro | - | - | 75.0 & 81.3 | - | - | 78.2 |
| VILA-40B | 54.0 | - | 60.1 & 61.1 | - | 67.9 | 60.8 |
| LLaVA-N-Video-32B | 59.4 | - | 60.2 & 63.0 | 65.5 | 77.3 | 65.1 |
| LLaVA-OneVision-7B | 57.1 | 58.4 | 58.6 & 61.8 | 64.8 | 79.3 | 63.3 |
| ST-VLM-7B (Ours) | **64.1** | **61.5** | **61.1** & **64.1** | **65.9** | **80.9** | **66.3** |

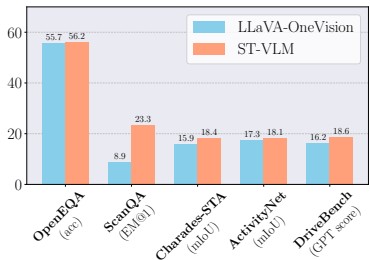

Figure 6: **Results on spatial and temporal reasoning benchmarks.**

Table 8: **Results on spatial & temporal categories.**

| Models | PerceptionTest physics | MVBench moving direction | VideoMME spatial perception | VideoMME temporal perception | LongVideoBench object tracking |
|---|---|---|---|---|---|
| LLaVA-OneVision | 49.0 | 21.5 | 57.4 | 63.6 | 43.8 |
| ST-VLM (Ours) | **58.5** | **35.0** | **66.7** | **76.4** | **47.9** |

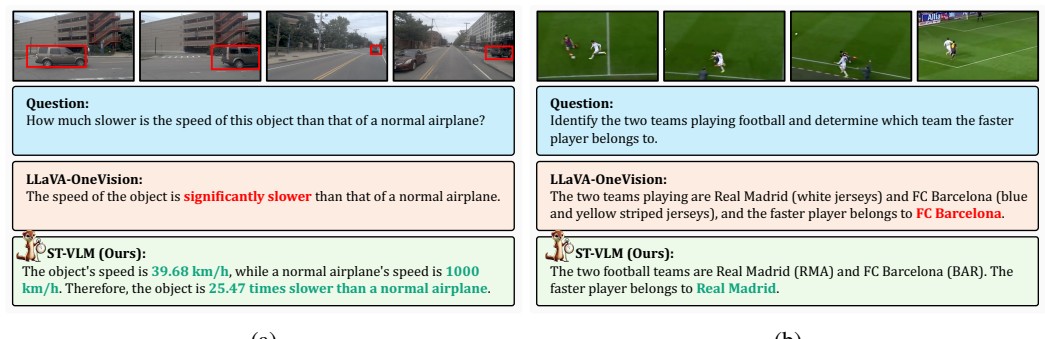

(a)          (b)

Figure 7: **Qualitative results on emergent capabilities of ST-VLM with multi-step reasoning.**

LLaVA-OneVision by 3.0% in average accuracy on general video understanding benchmarks. In particular, Tab. 8 reports results on spatial- and temporal-related categories. ST-VLM achieves substantial improvements over LLaVA-OneVision in spatio-temporal tasks, including moving direction, spatial/temporal perception, and object tracking, by effectively leveraging kinematic priors from STKit. Qualitative examples are provided in Sec. D.4.

Furthermore, Fig. 6 compares the performance of ST-VLM with LLaVA-OneVision on spatial and temporal reasoning benchmarks across diverse tasks and domains. For spatial reasoning, we evaluate on 3D scene understanding benchmarks OpenEQA (Majumdar et al., 2024) and ScanQA (Azuma et al., 2022), where ST-VLM improves performance by 0.5% and 14.4%, respectively. For temporal reasoning, ST-VLM surpasses LLaVA-OneVision by 2.5% and 0.8% mIoU on the video temporal grounding tasks of Charades-STA (Gao et al., 2017) and ActivityNet (Caba Heilbron et al., 2015). Finally, even on autonomous driving benchmarks that demand complex spatio-temporal reasoning, ST-VLM achieves a 2.4% gain over LLaVA-OneVision. These results demonstrate that incorporating the kinematics-based STKit dataset not only enhances general video understanding but also strengthens spatio-temporal reasoning across diverse scenarios.

## 6.3 EMERGENT CAPABILITIES OF ST-VLM

Finally, we answer **Q3** through qualitative analyses in Fig. 7 and 1b, showcasing ST-VLM's emergent multi-step reasoning capabilities that involve spatio-temporal reasoning. Although not explicitly trained for complex reasoning, ST-VLM effectively integrates kinematic reasoning with the existing abilities of VLMs, such as commonsense knowledge, logical inference, and arithmetic computation. For example, in Fig. 7a, when asked "How much slower is this object's speed compared to a normal airplane?", a model must (1) recall the average speed of a normal airplane, (2) estimate the object's speed from the video, and (3) perform arithmetic to compare them. Leveraging kinematic reasoning, ST-VLM produces an accurate answer (25.47 times slower), whereas LLaVA-OneVision provides a less precise response (10 times slower) without explicit reasoning. Similarly, in Fig. 7b, identifying the faster player requires recognizing teams by jersey color and estimating player speeds. ST-VLM correctly identifies the faster player as belonging to Real Madrid, whereas

the baseline fails to do so. These examples demonstrate the effectiveness of STKit-trained ST-VLM in enabling multi-step reasoning grounded in kinematics-based spatio-temporal understanding.

## 7 CONCLUSION

We present ST-VLM, a VLM with enhanced spatio-temporal reasoning capabilities, achieved through kinematic understanding in dynamic videos. To this end, we introduce STKit and STKit-Bench, which define seven fundamental tasks based on 3D-annotated video data. Furthermore, our 4D reconstruction-based data generation pipeline, along with the filtering strategy, effectively alleviates the scarcity of 3D annotations. Extensive analyses reveal that ST-VLM generalizes well across diverse video benchmarks and exhibits emergent multi-step reasoning by combining the pretrained knowledge of VLMs with newly acquired kinematic understanding.

## ETHICS STATEMENT

Our pseudo-labeling pipeline does not raise direct ethical concerns. However, the SFT datasets used for training ST-VLM may contain biases, such as those related to religion, gender, or race, which could lead ST-VLM to implicitly inherit these biases.

## REPRODUCIBILITY STATEMENT

The 4D reconstruction-based pseudo-labeling pipeline and the filtering strategy are described in Sec. 3.3. For reproducibility, their implementation details are presented in Secs. A.1 and B.1, respectively. Furthermore, the training details of ST-VLM are provided in Sec. A.2.

## THE USE OF LARGE LANGUAGE MODELS (LLMS)

We use LLMs for sentence-level refinement.

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

# APPENDIX

The appendix is organized into the following sections:

## A IMPLEMENTATION DETAILS

### A.1 DETAILS OF PSEUDO-LABELING PIPELINE

First, for the geometric reconstruction branch, we employ Monst3r (Zhang et al., 2025b) for 4D scene reconstruction on dynamic videos. In detail, we set the window size to 5 and use a scene graph configuration of swinstride-5-noncyclic to generate image pairs for feature matching. The reconstruction is performed with MonST3R using a temporal smoothing weight of 0.01, a translation weight of 1.0, and a flow loss weight of 0.01, applied after 10% of the total iterations and only to flow values exceeding a threshold of 25. This process runs for 300 iterations with a learning rate of 0.01 under a linear schedule. To address the scale misalignment issue, we canonicalize the reconstructed 4D scenes by rescaling MonST3R's depth estimates with the metric depths provided by Metric3D v2 (Hu et al., 2024).

Second, in the semantic reconstruction branch, we utilize Grounded-SAM2 (Ren et al., 2024) to extract bounding boxes, segmentation masks, and trajectories of selected objects. We focus on object categories related to dynamic movements, including "bus," "car," "vehicle," "human," "automobile," "person," "animal," "bicycle," "motorcycle," and "truck," which are provided to Grounded-SAM2 as text prompts. Overall, kinematic grounding in a canonicalized 4D scene requires approximately 400 seconds per video on a single A6000 GPU.

## A.2 DETAILS OF ST-VLM TRAINING

Our ST-VLM is initialized from LLaVA-OneVision 7B Li et al. (2024a) and trained with 63K STKit samples, 500K LLaVA-Video samples, 500K LLaVA-OneVision samples, and 100K OpenSpatial-Dataset samples. Training is performed on $8 \times$ A6000 GPUs for one epoch, taking approximately two weeks. We adopt a cosine learning rate scheduler with an initial learning rate of 1e-5 and a batch size of 128, using up to 32 frames per video for training and inference. For each video input, we provide additional temporal context in the form: "The video lasts for $t$ seconds, and $n$ frames are uniformly sampled from it. These frames are located at $t_1, t_2, \ldots, t_n$ seconds." For STKit samples, we further provide information about the visual prompt: "There are $k$ objects annotated with [COLOR] bounding boxes in the video."

## B DISCUSSION ON THE FILTERING STRATEGY

### B.1 DETAILS OF THE FILTERING STRATEGY

To ensure reliable centroid trajectory estimations in our pseudo-labeling pipeline, we develop a three-stage filtering strategy, calibrated by empirically comparing estimated trajectories against GT trajectories in the NuScenes dataset (Caesar et al., 2020), which provides LiDAR-annotated videos.

**Rule-based filtering.** We design heuristics to remove unreliably reconstructed scenes. Specifically, we eliminate noisy point clouds using DBSCAN (min_points = 5), discard detections with box confidence below 0.4 or text confidence below 0.3, and exclude bounding boxes with an area smaller than 10 pixels. After kinematic grounding, we detect trajectory outliers by removing centroid coordinates with a Z-score above 3.0 or a cosine similarity below -0.2 relative to the mean direction vector to discard trajectories containing such outliers. We then reorder each trajectory using the nearest neighbor algorithm to enforce spatio-temporal consistency, followed by smoothing with a 3D Kalman filter (process_variance = 1.0, measurement_variance = 1000). These hyperparameters are selected by comparison with GT trajectories on NuScenes, and subsequently applied during pseudo-labeling to produce more accurate labels for unlabeled videos.

**General model-based filtering.** We employ a VLM to filter out scenes with occluded objects, significant camera motion, or poor object detection and tracking. Specifically, LLaVA-OneVision, the initialization for our ST-VLM, is used to assess these criteria based on the prompt in Tab. 9. Scenes that do not satisfy any of these criteria are discarded.

Table 9: **Prompts used for VLMs in general model-based filtering.**

> `Occlusion:` Is the object inside each bounding box fully visible, without significant occlusion? Respond with 'Yes' or 'No'.
>
> `Camera movement:` Do the video frames transition smoothly, without noticeable temporal discontinuities? Respond with 'Yes' or 'No'.
>
> `Object detection:` Is each bounding box tightly enclosing an individual object, without significant misalignment or cropping? Respond with 'Yes' or 'No'.
>
> `Object tracking:` Does each bounding box reliably track the target object across all frames, without losing alignment or missing the object? Respond with 'Yes' or 'No'

**Task-specific model-based filtering.** In this stage, we utilize a model trained only on 3D-annotated datasets, *i.e.*, NuScenes (Caesar et al., 2020), Argoverse2 (Wilson et al., 2023), and Ego-Exo4D (Grauman et al., 2024), to filter out low-quality pseudo-labeled samples based on likelihood scores. For each sample, the model computes a likelihood, which is then normalized using min-max scaling within the same task across the seven defined tasks. We apply task-specific thresholds: 0.8 for Traveled Distance and Traveling Speed, 0.4 for Traveled Distance Comparison and Traveling Speed Comparison, and 0.7 for Movement Direction and Movement Direction Comparison. Samples with likelihood scores below these thresholds are discarded.

## B.2 ANALYSIS ON THE FILTERING STRATEGY

We provide an in-depth analysis to verify the effectiveness of our filtering strategy. Tab. 10 presents an ablation study, reporting the number of low-quality samples and the average accuracy on STKit-Bench at each filtering stage. Low-quality samples are defined as those that fail to meet the criteria in Sec. B.1, as evaluated by the advanced VLM InternVL3 (Zhu et al., 2025). The results indicate that our filtering strategy significantly reduces the number of low-quality samples. For example, the number of samples with occluded objects drops from 12,035 to 174 after the three-stage filtering. This reduction leads to a notable performance improvement on STKit-Bench, increasing accuracy from 40.4% to 46.1%.

Table 10: **Ablation study on the filtering strategy.**

| Filtering strategy | | | Number of low-quality samples ↓ | | | | avg. acc.↑ |
|---|---|---|---|---|---|---|---|
| Rule-based | General model-based | Task-specific model-based | Occlusion | Camera movement | Object detection | Object tracking | |
| - | - | - | 12,035 | 28 | 9,278 | 118,856 | 40.4 |
| ✔ | - | - | 2,123 | 4 | 1,463 | 18,274 | 41.0 |
| ✔ | ✔ | - | 348 | 4 | 419 | 9,905 | 41.9 |
| ✔ | ✔ | ✔ | **174** | **3** | **206** | **5,962** | **46.1** |

We further compare the computed trajectories against GT trajectories from the 3D-annotated dataset, NuScenes. For Traveled Distance, the mean error rate decreases from 207% to 29% after applying the three-stage filtering, where the error rate is defined as $\frac{|\text{Pred}-\text{GT}|}{\text{GT}}$. These results demonstrate that the filtering strategy substantially improves pseudo-label quality and overall performance.

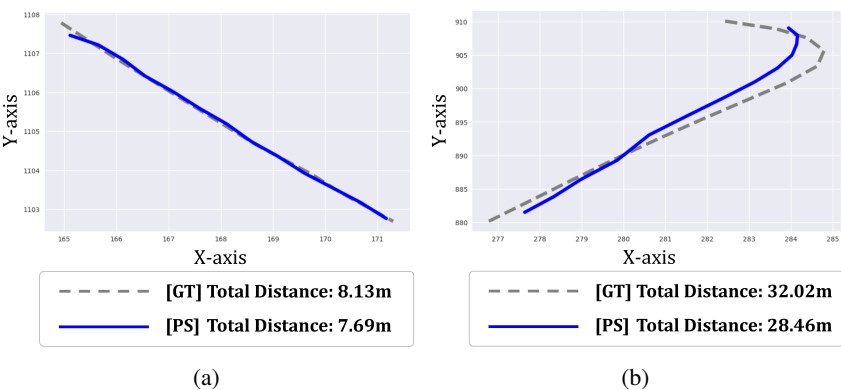

(a)                                                                (b)

Figure 8: **Comparison of projected trajectories.** GT trajectory is shown in gray dash line and estimated trajectory from pseudo-label (PS) is shown in blue solid line.

Fig. 8 compares GT trajectories, shown as dashed gray lines, with our estimated trajectories, shown as solid blue lines. The alignment indicates close correspondence in straight-line movements (Fig. 8a), which are common in real-world scenarios, with only minor deviations in curved paths (Fig. 8b). Quantitatively, the estimated traveled distances show only small deviations from the GT distances, *e.g.*, with an error of 0.44 m for straight-line movements and 3.56 m for curved paths. These results highlight the reliability of our pseudo-labeling pipeline for estimating object trajectories without requiring 3D annotations.

## C FURTHER QUANTITATIVE RESULTS

### C.1 RESULTS ON VARIOUS SCENARIOS OF STKIT-BENCH

Tabs. 11–15 demonstrate the results of ST-VLM across diverse scenarios in STKit-Bench. These results highlight the robustness of ST-VLM despite challenges such as object occlusion, multi-object scenarios, dynamic scenes, small object sizes, and varying FPS.

Table 11

| occlusion | single object | multiple objects | avg. acc. |
|---|---|---|---|
| not occluded | 46.9% (363 / 774) | 74.3% (350 / 471) | 57.3% (713 / 1,245) |
| occluded | 26.9% (7 / 26) | 73.6% (95 / 129) | 65.8% (102 / 155) |

Table 12

| number of objects | single object | multiple objects | avg. acc. |
|---|---|---|---|
| few | 52.0% (204 / 392) | 73.8% (236 / 320) | 61.8% (440 / 712) |
| several | 42.1% (130 / 309) | 76.4% (162 / 212) | 56.1% (292 / 521) |
| many | 36.4% (36 / 99) | 69.1% (47 / 68) | 49.7% (83 / 167) |

Table 13

| camera movement | single object | multiple objects | avg. acc. |
|---|---|---|---|
| static | 46.1% (360 / 781) | 74.5% (444 / 596) | 58.4% (804 / 1,377) |
| dynamic | 43.8% (7 / 16) | 57.1% (4 / 7) | 47.8% (11 / 23) |

Table 14

| object size | single object | multiple objects | avg. acc. |
|---|---|---|---|
| small | 60.3% (38 / 63) | 67.7% (88 / 130) | 65.3% (440 / 712) |
| medium | 65.6% (88 / 187) | 76.6% (242 / 316) | 56.1% (292 / 521) |
| large | 44.4% (244 / 550) | 74.7% (115 / 154) | 49.7% (83 / 167) |

Table 15

| FPS | avg. acc. |
|---|---|
| 2 | 58.2 |
| 1 | 56.8 |
| 0.5 | 46.3 |
| 0.25 | 40.2 |

Table 16: **Results on each domain of STKit-Bench.** AD stands for autonomous driving.

| Train | Test | Traveled Distance | | Traveling Speed | |
|---|---|---|---|---|---|
| | | Acc↑ | MAE↓ | Acc↑ | MAE↓ |
| - | Sports | 16.0 | 6.7 | 8.0 | 2.8 |
| AD | Sports | 22.0 | 4.8 | 64.0 | 0.9 |
| Sports | Sports | **76.0** | 2.0 | **78.0** | 0.8 |
| AD + Sports | Sports | **76.0** | **1.7** | **78.0** | **0.7** |
| - | AD | 13.0 | 49.1 | 10.0 | 24.3 |
| AD | AD | 35.5 | 21.7 | 32.0 | **12.6** |
| Sports | AD | 5.0 | 36.4 | 6.5 | 29.0 |
| AD + Sports | AD | **38.5** | **17.0** | **32.5** | **12.6** |

## C.2 RESULTS ON EACH DOMAIN OF STKIT-BENCH

Tab. 16 presents cross-domain evaluation between autonomous driving and sports. Training solely on autonomous driving data substantially improves performance in the sports domain. For instance, in Traveling Speed, accuracy increases from 8.0% to 64.0%. In contrast, training only on sports data provides no improvement for autonomous driving. We attribute this to the difficulty of learning vehicle motion patterns, such as traveled distance and speed, from the relatively limited sports data in STKit. By incorporating both domains, the model achieves the best performance, underscoring the importance of broad domain coverage.

## C.3 RESULTS ON OUT-OF-DOMAIN SETTINGS

Table 17: **Results on out-of-domain settings.**

| | in-domain (NuScenes & Argoverse2) | out-of-domain (NuPlan) | out-of-domain (Waymo) |
|---|---|---|---|
| LLaVA-OneVision (Li et al., 2024a) | 35.4 | 26.0 | 30.8 |
| ST-VLM (**ours**) | **63.4** (+28.0) | **55.6** (+29.6) | **58.4** (+27.6) |

To evaluate out-of-domain generalization, STKit-Bench incorporates NuPlan data, which is not included in the training set, constituting 74.8% of the evaluation benchmark and featuring distinct camera intrinsics/extrinsics, road scenes, weather conditions, illumination, and locations. We further conduct an additional evaluation on the Waymo dataset (Sun et al., 2020), which also employs different camera parameters at different road scenes. Tab. 17 shows detailed results across in-domain and out-of-domain settings, demonstrating the robustness of our model to variations in camera configurations and road scenes across datasets.

## C.4 RESULTS OF SIMULATION DATA

Table 18: **Results of simulation data.**

|  | Accuracy |
|---|---|
| LLaVA-OneVision (Li et al., 2024a) | 26.4 |
| ST-VLM (simulation) | 29.1 |
| ST-VLM (pseudo) | 46.1 |
| ST-VLM (GT) | **52.6** |

We construct kinematic instruction data using two simulation datasets, VKITTI (Gaidon et al., 2016) and GTA V (Richter et al., 2017), to evaluate the effectiveness of simulation-based videos for real-world evaluation scenarios. Tab. 18 reports the performance of ST-VLM trained with simulation data, showing that simulation alone provides performance gains, although the improvement is smaller compared to training with real-world videos, *i.e.*, pseudo-labeled and GT-labeled data, due to the domain gap.

## C.5 RESULTS ON DEPTH ESTIMATION

Table 19: **Results on depth estimation.**

|  | STKit-Bench (acc.) | DepthLMBench (acc. / MAE) |
|---|---|---|
| LLaVA-OneVision (Li et al., 2024a) | 26.4 | 7.5 / 45.9 |
| DepthLM (Cai et al., 2025b) | 13.2 | 19.7 / **9.9** |
| ST-VLM (**ours**) | **58.2** | **21.2** / 10.2 |

Surprisingly, our kinematic instruction tuning enables ST-VLM to implicitly acquire depth understanding as part of its kinematic understanding process, even though our dataset does not contain any explicit depth estimation samples. Tab. 19 compares the performance on depth estimation with DepthLM (Cai et al., 2025b), which is a specialized model dedicated solely to depth estimation and does not generalize to kinematic understanding tasks. ST-VLM achieves 21.2% accuracy and a 10.2 m MAE on DepthLMBench despite no explicit depth-specific supervision, whereas DepthLM cannot estimate the object kinematic quantities required in STKit-Bench, underscoring the broader reasoning capability of our model.

## D FURTHER QUALITATIVE RESULTS

### D.1 RESULTS ON STKIT-BENCH

We present qualitative results on STKit-Bench, comparing ST-VLM with baseline models such as GPT-5 and LLaVA-OneVision (Li et al., 2024a). Fig. 15 and 16 illustrate examples across seven spatio-temporal reasoning tasks: traveled distance, traveling speed, movement direction, direction timestamp, traveled distance comparison, traveling speed comparison, and movement direction comparison.

### D.2 RESULTS ON CHALLENGING SAMPLES

In Fig. 17, we present additional qualitative results on challenging cases. As shown in Fig. 17a and 17b, ST-VLM successfully predicts the traveled distance of objects with complex trajectories. In Fig. 17d, we further assess the model's spatio-temporal reasoning ability without visual prompts by removing bounding boxes and providing only textual instructions with object attributes. For the query, "Which is moving slower, the cyclist on the right or the yellow trailer?", LLaVA-OneVision incorrectly predicts that the cyclist is moving slower than the yellow trailer, whereas ST-VLM correctly identifies the yellow trailer as slower. This demonstrates that ST-VLM leverages video evidence to answer accurately, while LLaVA-OneVision tends to rely on commonsense priors (*e.g.*, vehicles are generally faster than bicycles), leading to erroneous predictions.

### D.3 RESULTS OF EMERGENT CAPABILITIES

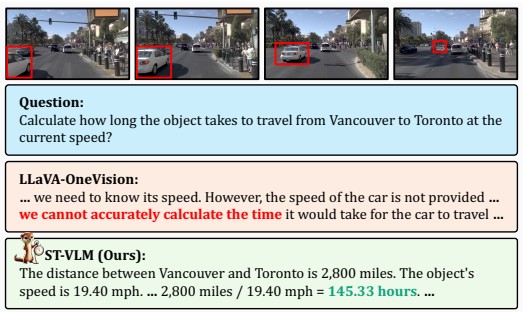

Figure 9: **Qualitative results of emergent capabilities.**

Fig. 9 presents an example of a complex instruction that requires multi-step reasoning. The task involves integrating spatio-temporal understanding with existing capabilities (*e.g.*, commonsense knowledge, logical reasoning, and arithmetic computation). Our ST-VLM successfully derives the correct answer, whereas LLaVA-OneVision fails.

### D.4 RESULTS ON PERCEPTIONTEST

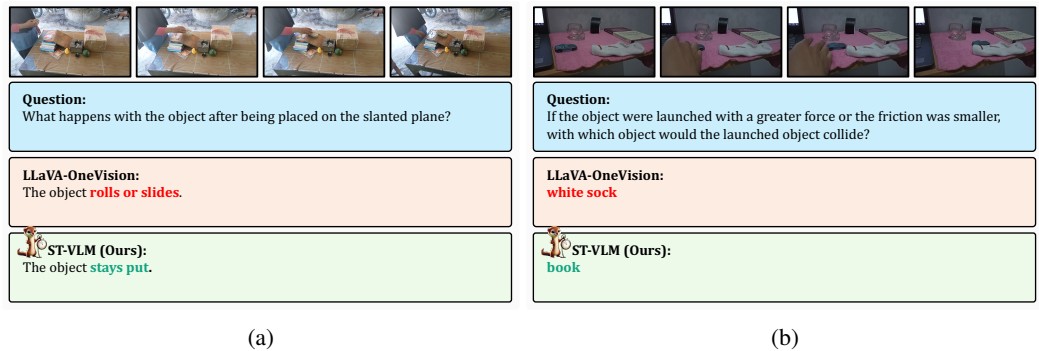

(a)                                                                          (b)

Figure 10: **Qualitative results on PerceptionTest.**

Fig. 10 provides qualitative results of ST-VLM on PerceptionTest (Pătrăucean et al., 2023). In Fig. 10a, ST-VLM correctly answers "The object stays put" to the question, "What happens with the object after being placed on the slanted plane?", demonstrating its ability to reason over learned object kinematics. In contrast, LLaVA-OneVision predicts the incorrect answer, "The object rolls or slides," likely due to over-reliance on textual cues (e.g., "placed on slanted plane") rather than visual reasoning.

### D.5 RESULTS OF GPT-5

Fig. 11 provides a qualitative example of GPT-5 with additional context, *i.e.*, in-context examples, depth maps, and camera extrinsics. We observe that GPT-5 often replicates the GT (2.45 m) from in-context examples rather than engaging in genuine reasoning.

### D.6 RESULTS ON EXTRAORDINARY SCENARIOS

We observe strong generalization even in extraordinary scenarios involving a remote-controlled (RC) car, which we attribute to the pseudo-labeled data sourced from diverse domains beyond road scenes. As illustrated in Fig. 12, when asked "Can you calculate the total distance covered by the red RC car throughout the entire video?", ST-VLM estimates the traveled distance as 12.73 m, which lies within a plausible range for the actual distance.

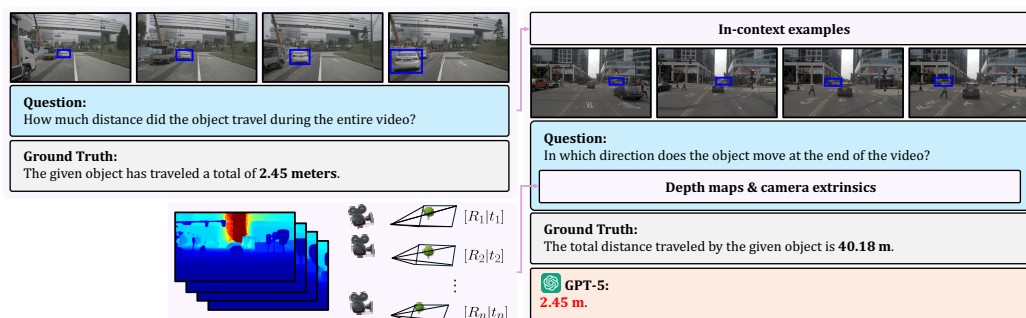

Figure 11: **Qualitative results of GPT-5.** Even with additional context, we observe that GPT-5 tends to *replicate* the GT of **in-context examples** rather than performing genuine reasoning.

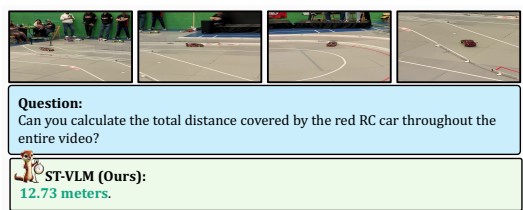

Figure 12: **Qualitative results of extraordinary scenarios.**

# E DETAILS OF STKIT-BENCH

## E.1 COMPARISON WITH OTHER BENCHMARKS

Recently, several video benchmarks have been proposed in the literature (Li et al., 2024b; Pătrăucean et al., 2023; Fu et al., 2024; Zhou et al., 2024a). For example, MLVU (Zhou et al., 2024a) aims to assess video-based VLMs for long-form video understanding, and VideoMME (Fu et al., 2024) focuses on the comprehensive perception ability of the model on a wide range of domains. To tackle the problem that most VLMs overlook the temporal information, MVBench (Li et al., 2024b) has been proposed by covering diverse temporal understanding tasks, *e.g.*, action sequence understanding, action prediction, and counterfactual inference. More recently, several works (Wang et al., 2024b; Zhou et al., 2024b; Ding et al., 2024; Nie et al., 2024) have been introduced as a video spatio-temporal understanding benchmark for autonomous driving scenes. In contrast, our STKit-Bench covers general scenes, *e.g.*, sports, not limited to autonomous driving scenarios.

## E.2 STATISTICS

Fig. 13 illustrates the statistics of STKit-Bench. Directly adopting the generated QA pairs for benchmarking results in a long-tail label distribution, which we mitigate by balancing the labels, as shown in Fig. 13a and 13b. The red bars highlight the imbalanced distribution in both distance and direction categories, while the green bars indicate the balanced distribution. Fig. 13c illustrates the dataset composition: 74.8% NuPlan (Caesar et al., 2021), 12.5% NuScenes (Caesar et al., 2020), 5.6% Argoverse2 (Wilson et al., 2023), and 7.1% Ego-Exo4D (Grauman et al., 2024). We primarily use NuPlan, which is not included in the training data, to evaluate out-of-domain scenarios in STKit-Bench.

## E.3 GPT-5-NANO PROMPTS FOR EVALUATION

Tab. 20-26 present the prompts used with GPT-5-nano for evaluation on STKit-Bench. During evaluation, our goal is to extract only the essential information from the final VLM outputs. To this end, we convert the outputs into JSON format using GPT-5-nano with the designed prompts. These JSON files are then used for the final task evaluation, as detailed in Sec. 4.

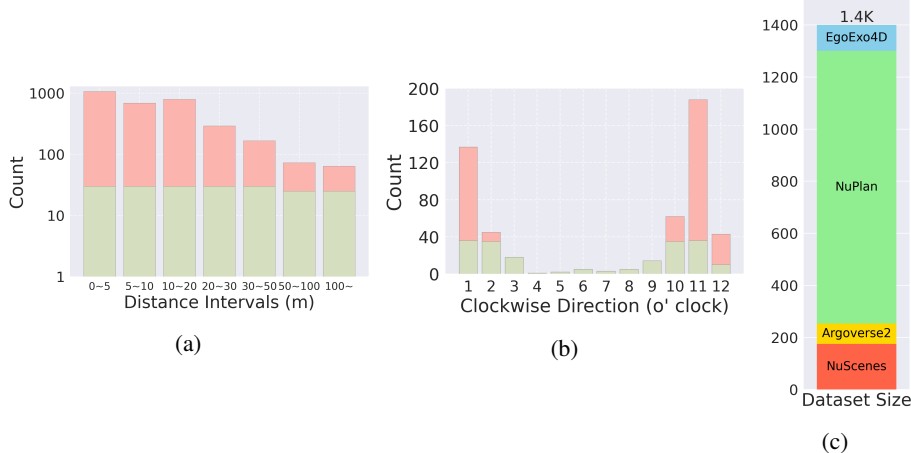

(a)

(b)

(c)

Figure 13: **Statistics of STKit-Bench.** (a), (b) We balance the number of samples for each label to prevent biased results. Red and green bars indicate the number of samples before/after balancing. (c) shows the composition of STKit-Bench.

## F  QA TEMPLATES FOR STKIT

Tab. 27–36 provide all the QA templates used in STKit, as detailed in Sec. 3.4. These templates are designed as kinematic instructions for each spatio-temporal reasoning task.

## G  EXAMPLES OF FILTERING

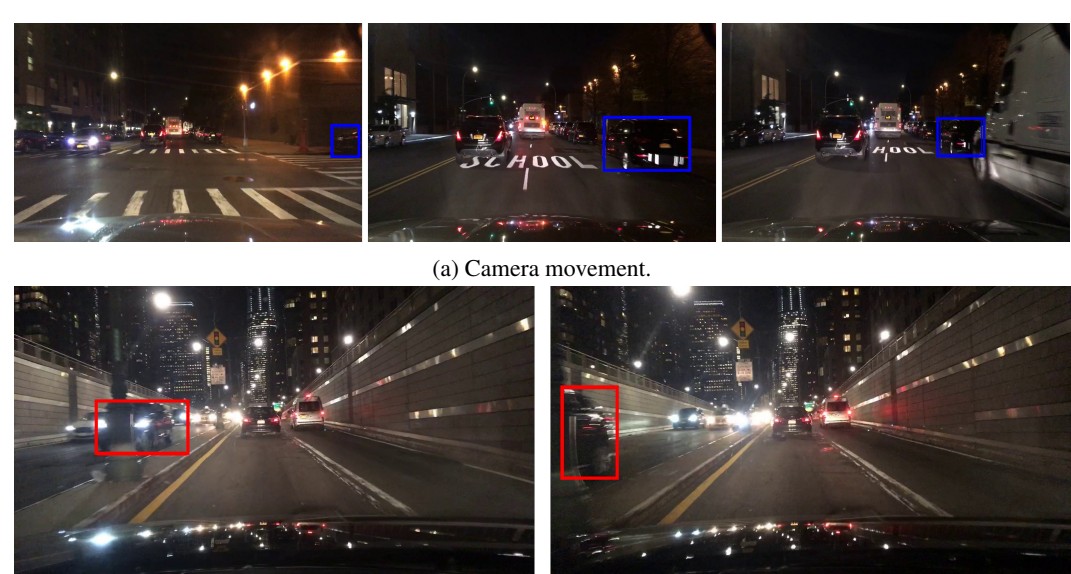

(a) Camera movement.

(b) Occlusion.

Figure 14: **Examples of filtered samples.**

To illustrate how the model identifies low-quality samples, we provide examples of filtered cases in Fig. 14. In Fig. 14a, the model detects a significant scene transition between the first and second frames. In Fig. 14b, it successfully identifies object occlusion, demonstrating that our filtering strategy effectively removes low-quality samples. Overall, applying this filtering strategy yields a 5.7% performance improvement.

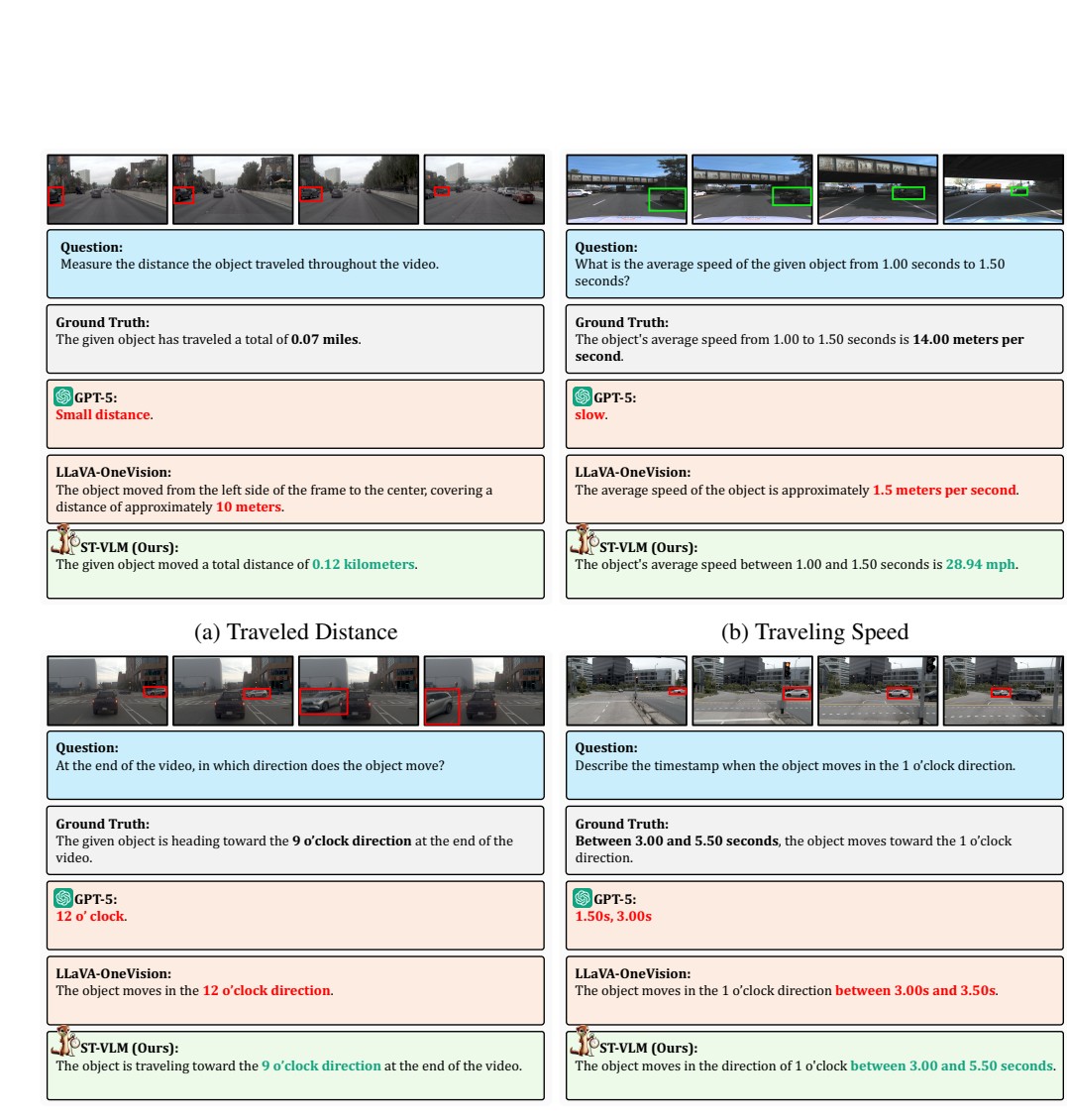

Figure 15: **Qualitative results on STKit-Bench.**

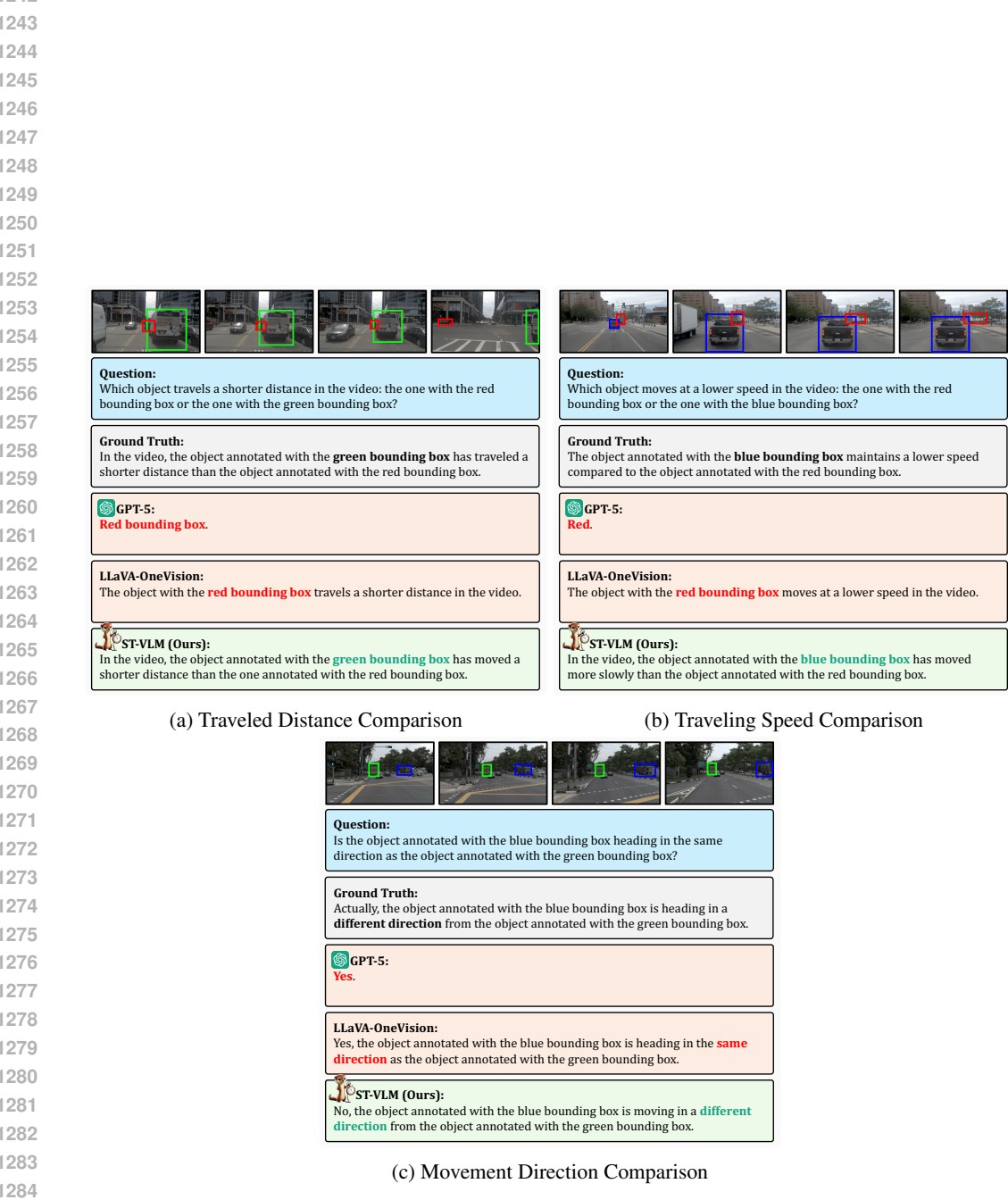

(a) Traveled Distance Comparison

(b) Traveling Speed Comparison

(c) Movement Direction Comparison

Figure 16: **Qualitative results on STKit-Bench.**

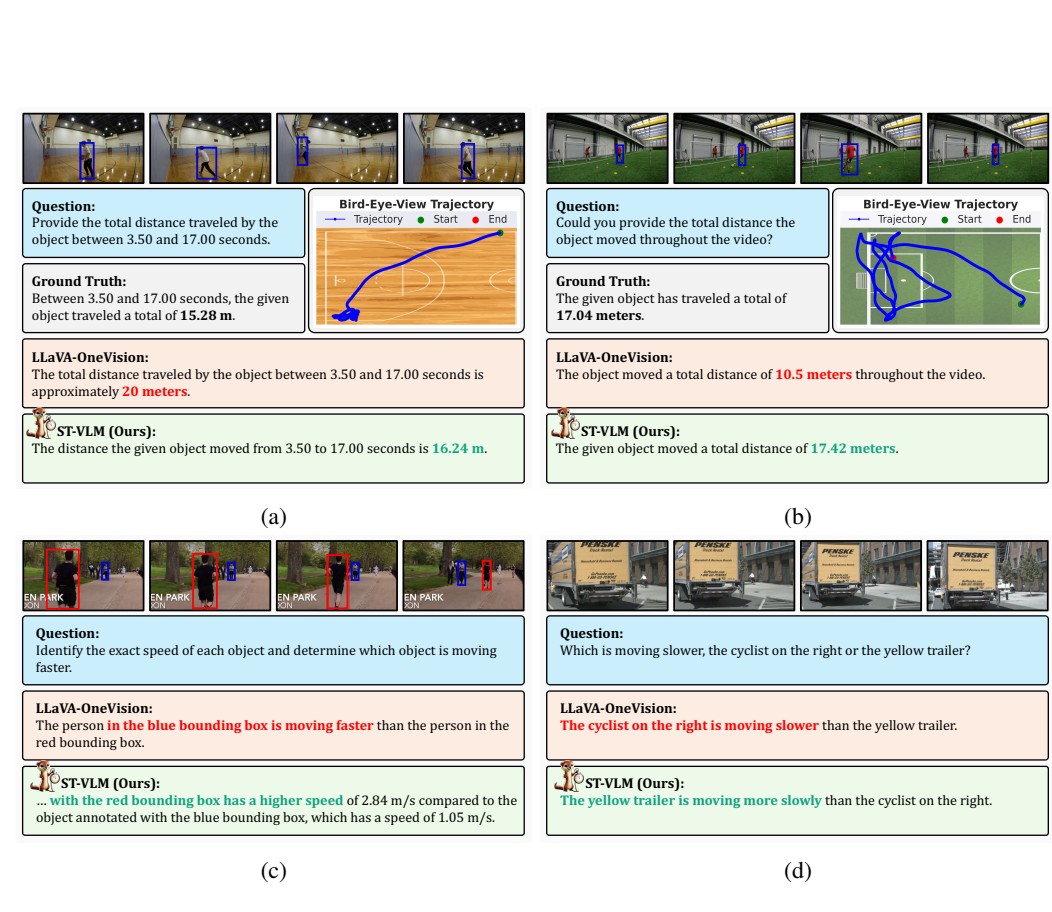

Figure 17: **Qualitative results on challenging examples.**

Table 20: **GPT-5-nano prompts for Traveled Distance.**

```
traveled_distance_prompt = f"""You should help me to evaluate the response given the
question and the correct answer.
You need to convert the distance of the correct answer and response to meters.
The conversion factors are as follows: 1 inch = 0.0254 meters. 1 foot = 0.3048 meters. 1 centimeter
(cm) = 0.01 meters.
You should output two floats in meters, one for the answer, and one for the response.
The output should be in JSON format."""

messages = [{"role":"system", "content":traveled_distance_prompt}]
for sample in fewshot_samples:
    messages.append({"role":"user", "content":sample['context']})
    messages.append({"role":"assistant", "content":sample['response']})
messages.append({"role":"user", "content":'\n'.join(query)})
```

Table 21: **GPT-5-nano prompts for Traveling Speed.**

```
traveling_speed_prompt = f"""You should help me to evaluate the response given the
question and the correct answer.
You need to convert the speed of the correct answer and response to kilometers per hour (km/h).
The conversion factors are as follows: 1 meters per second (m/s) = 3.6 kilometers per hour (km/h).
1 miles per hour (mph) = 1.60934 kilometers per hour (km/h). 1 foot per second (ft/s) = 1.09728
kilemoeters per hour (km/h).
You should output two floats in kilometers per hour (km/h), one for the answer, and one for the
response.
The output should be in JSON format."""

messages = [{"role":"system", "content":traveling_speed_prompt}]
for sample in fewshot_samples:
    messages.append({"role":"user", "content":sample['context']})
    messages.append({"role":"assistant", "content":sample['response']})
messages.append({"role":"user", "content":'\n'.join(query)})
```

Table 22: **GPT-5-nano prompts for Movement Direction.**

```
movement_direction_prompt = f"""You should help me to evaluate the response given
the question and the correct answer.
You need to extract the direction of the correct answer and response.
You should output two integers in clock directions, one for the answer, and one for the response.
The output should be in JSON format."""

messages = [{"role":"system", "content":movement_direction_prompt}]
for sample in fewshot_samples:
    messages.append({"role":"user", "content":sample['context']})
    messages.append({"role":"assistant", "content":sample['response']})
messages.append({"role":"user", "content":'\n'.join(query)})
```

Table 23: **GPT-5-nano prompts for Direction Timestamp.**

```
direction_timestamp_prompt = f"""You should help me to evaluate the response given
the question and the correct answer.
You need to extract the start time and end time in seconds of the correct answer and response.
You should output four floats in seconds, one for the answer start time, one for the answer end time,
one for the response start time, and one for the response end time.
The output should be in JSON format."""

messages =[{"role":"system", "content":direction_timestamp_prompt}]
for sample in fewshot_samples:
    messages.append({"role":"user", "content":sample['context']})
    messages.append({"role":"assistant", "content":sample['response']})

messages.append({"role":"user", "content":'\n'.join(query)})
```

Table 24: **GPT-5-nano prompts for Traveled Distance Comparison.**

```
distance_comparison_prompt = f"""You should help me to evaluate the response given
the question and the correct answer.
To mark a response, you should output a single integer between 0 and 1.
1 means that the response perfectly matches the answer.
0 means that the response is completely different from the answer.
The output should be in JSON format."""

messages =[{"role":"system", "content":distance_comparison_prompt}]
for sample in fewshot_samples:
    messages.append({"role":"user", "content":sample['context']})
    messages.append({"role":"assistant", "content":sample['response']})

messages.append({"role":"user", "content":'\n'.join(query)})
```

Table 25: **GPT-5-nano prompts for Traveling Speed Comparison.**

```
speed_comparison_prompt = f"""You should help me to evaluate the response given the
question and the correct answer.
To mark a response, you should output a single integer between 0 and 1.
1 means that the response perfectly matches the answer.
0 means that the response is completely different from the answer.
The output should be in JSON format."""

messages =[{"role":"system", "content":speed_comparison_prompt}]
for sample in fewshot_samples:
    messages.append({"role":"user", "content":sample['context']})
    messages.append({"role":"assistant", "content":sample['response']})

messages.append({"role":"user", "content":'\n'.join(query)})
```

Table 26: **GPT-5-nano prompts for Movement Direction Comparison.**

```python
direction_comparison_prompt = f"""You should help me to evaluate the response given
the question and the correct answer.
To mark a response, you should output a single integer between 0 and 1.
1 means that the response perfectly matches the answer.
0 means that the response is completely different from the answer.
The output should be in JSON format."""

messages = [{"role":"system", "content":direction_comparison_prompt}]
for sample in fewshot_samples:
    messages.append({"role":"user", "content":sample['context']})
    messages.append({"role":"assistant", "content":sample['response']})

messages.append({"role":"user", "content":'\n'.join(query)})
```

Table 27: **QA templates for Traveled Distance.**

```
traveled_distance_common_prompt = f"""The video lasts for [SECONDS] seconds,
and [FRAMES] frames are uniformly sampled from it. These frames are located at [SEC-
OND1]s,[SECOND2]s,[SECOND3]s, ... . Please answer the following questions related to this
video.

There is an object annotated with a [COLOR] bounding box in the video. """

traveled_distance_questions = [
        "What is the total distance traveled by the given object in the video?",
        "Can you calculate the total distance covered by the object in the video?",
        "Tell me the overall distance the object has traveled in the video.",
        "Could you provide the total distance the object moved throughout the video?",
        "How much distance did the object travel during the entire video?",
        "Measure the distance the object traveled throughout the video.",
        "What is the total distance traveled by a given object from [START] seconds to
        [END] seconds?",
        "Can you calculate the total distance the object traveled between [START] sec-
        onds and [END] seconds?",
        "Tell me how far the object moved from [START] seconds to [END] seconds.",
        "Could you measure the total distance the object covered between [START]
        and [END] seconds?",
        "How much distance did the object travel during the period from [START] to
        [END] seconds?",
        "Provide the total distance traveled by the object between [START] and [END]
        seconds."
]

traveled_distance_answers = [
        "The total distance traveled by the given object is [DISTANCE].",
        "The given object's traveled distance is [DISTANCE].",
        "The given object has traveled a total of [DISTANCE].",
        "The entire distance the given object traveled amounts to [DISTANCE].",
        "The given object moved a total distance of [DISTANCE].",
        "The distance traveled by the given object from [START] to [END] seconds is
        [DISTANCE].",
        "The given object traveled [DISTANCE] between [START] and [END] sec-
        onds.",
        "From [START] to [END] seconds, the given object moved a distance of [DIS-
        TANCE].",
        "The distance the given object moved from [START] to [END] seconds is [DIS-
        TANCE].",
        "Between [START] and [END] seconds, the given object traveled a total of
        [DISTANCE]."
]
```

Table 28: **QA templates for Traveling Speed.**

```
traveling_speed_common_prompt = f"""The video lasts for [SECONDS] seconds,
and [FRAMES] frames are uniformly sampled from it. These frames are located at [SEC-
OND1]s,[SECOND2]s,[SECOND3]s, ... . Please answer the following questions related to this
video.

There is an object annotated with a [COLOR] bounding box in the video. """

traveling_speed_questions = [
        "What is the average speed of the given object in the video?",
        "Calculate the average speed of the object in the video.",
        "Tell me the object's average speed throughout the video.",
        "Could you provide the average velocity of the object throughout the video?",
        "What is the object's average speed during the video?",
        "Can you measure the average speed for the object in the entire video?",
        "What is the average speed of the given object from [START] seconds to [END]
        seconds?",
        "Can you calculate the average speed of the object between [START] and
        [END] seconds?",
        "Tell me the object's average speed from [START] to [END] seconds.",
        "Could you provide the average velocity of the object during the time period
        from [START] to [END] seconds?",
        "What is the average speed of the object between [START] and [END] sec-
        onds?",
        "Measure the object's average velocity during the interval from [START] to
        [END] seconds?"
]

traveling_speed_answers = [
        "The average speed of the given object is [SPEED] throughout the video.",
        "The object's average speed across the entire video is [SPEED].",
        "Throughout the video, the object moves at an average speed of [SPEED].",
        "The given object maintains an average speed of [SPEED] during the entire
        video.",
        "The average velocity of the given object throughout the video is [SPEED].",
        "The average speed of the given object from [START] to [END] seconds is
        [SPEED].",
        "The object's average speed between [START] and [END] seconds is
        [SPEED].",
        "From [START] to [END] seconds, the object moves at an average speed of
        [SPEED].",
        "The given object has an average velocity of [SPEED] during the time period
        from [START] to [END] seconds.",
        "The object's average speed from [START] to [END] seconds is [SPEED]."
]
```

Table 29: **QA templates for Movement Direction.**

```
movement_direction_common_prompt = f"""The video lasts for [SECONDS] seconds,
and [FRAMES] frames are uniformly sampled from it. These frames are located at [SEC-
OND1]s,[SECOND2]s,[SECOND3]s, ... . Please answer the following questions related to this
video.

There is an object annotated with a [COLOR] bounding box in the video. """

movement_direction_questions = [
        "In which direction does the object move at the end of the video?",
        "What direction does the object travel at the end of the video?",
        "At the end of the video, in which direction does the object move?",
        "Describe the direction of the object moving at the end of the video.",
        "Provide the direction of the object moving at the end of the video."
]

movement_direction_answers = [
        "The given object is heading toward the [CLOCK] o'clock direction at the end
        of the video.",
        "The object moves in the direction of [CLOCK] o'clock at the end of the
        video.",
        "At the end of the video, the object is heading toward the [CLOCK] o'clock
        direction.",
        "The object is traveling toward the [CLOCK] o'clock direction at the end of
        the video.",
        "At the end of the video, the object moves toward the [CLOCK] o'clock direc-
        tion."
]
```

Table 30: **QA templates for Direction Timestamp.**

```
direction_timestamp_common_prompt = f"""The video lasts for [SECONDS] seconds,
and [FRAMES] frames are uniformly sampled from it. These frames are located at [SEC-
OND1]s,[SECOND2]s,[SECOND3]s, ... . Please answer the following questions related to this
video.

There is an object annotated with a [COLOR] bounding box in the video. """

direction_timestamp_questions = [
        "Describe the timestamp when the object moves in the [CLOCK] o'clock di-
        rection.",
        "Can you provide the moment when the object moves in the [CLOCK] o'clock
        direction?",
        "Explain the time at which the object heads toward the [CLOCK] o'clock di-
        rection.",
        "At what timestamp does the object start moving in the [CLOCK] o'clock di-
        rection?",
        "When does the object begin traveling in the [CLOCK] o'clock direction?"
]

direction_timestamp_answers = [
        "The given object is heading toward the [CLOCK] o'clock direction from
        [START] to [END] seconds.",
        "The object moves in the direction of [CLOCK] o'clock between [START] and
        [END] seconds.",
        "From [START] to [END] seconds, the object is heading toward the [CLOCK]
        o'clock direction.",
        "The object is traveling toward the [CLOCK] o'clock direction during the time
        period from [START] to [END] seconds.",
        "Between [START] and [END] seconds, the object moves toward the [CLOCK]
        o'clock direction."
]
```

Table 31: **Question templates for Traveled Distance Comparison.**

```
traveled_distance_comparison_common_prompt = f"""The video lasts for [SEC-
ONDS] seconds, and [FRAMES] frames are uniformly sampled from it. These frames are located
at [SECOND1]s,[SECOND2]s,[SECOND3]s, ... . Please answer the following questions related to
this video.

There are two objects annotated with [COLOR1] and [COLOR2] bounding boxes in the video. """

traveled_distance_comparison_positive_questions = [
        "Which object travels a greater distance in the video: the one with the
        [COLOR1] bounding box or the one with the [COLOR2] bounding box?",
        "Which object moves farther throughout the video, the object annotated with
        the [COLOR1] bounding box or the [COLOR2] bounding box?",
        "Which object covers more distance in the video: the one annotated with the
        [COLOR1] bounding box or the [COLOR2] bounding box?",
        "Between the objects annotated with the [COLOR1] and [COLOR2] bounding
        boxes, which one moves a longer distance throughout the video?",
        "Between the two objects, one annotated with the [COLOR1] bounding box
        and the other annotated with the [COLOR2] bounding box, which one moves
        farther during the entire video?",
        "Which object, the one annotated with the [COLOR1] bounding box or the
        [COLOR2] bounding box, has a greater travel distance in the video?"
]

traveled_distance_comparison_negative_questions = [
        "Which object travels a shorter distance in the video: the one with the
        [COLOR1] bounding box or the one with the [COLOR2] bounding box?",
        "Which object moves a shorter distance throughout the video, the object anno-
        tated with the [COLOR1] bounding box or the [COLOR2] bounding box?",
        "Which object covers less distance in the video: the one annotated with the
        [COLOR1] bounding box or the [COLOR2] bounding box?",
        "Between the objects annotated with the [COLOR1] and [COLOR2] bounding
        boxes, which one moves a shorter distance throughout the video?",
        "Between the two objects, one annotated with the [COLOR1] bounding box
        and the other annotated with the [COLOR2] bounding box, which one moves
        less during the entire video?",
        "Which object, the one annotated with the [COLOR1] bounding box or the
        [COLOR2] bounding box, has a shorter travel distance in the video?"
]
```

Table 32: **Answer templates for Traveled Distance Comparison.**

```
traveled_distance_comparison_positive_answers = [
        "The object annotated with the [COLOR1] bounding box has traveled a greater
        distance than the object annotated with the [COLOR2] bounding box through-
        out the video.",
        "In the video, the object annotated with the [COLOR1] bounding box has
        moved a greater distance than the one annotated with the [COLOR2] bound-
        ing box.",
        "The object annotated with the [COLOR1] bounding box covers more distance
        throughout the video compared to the object annotated with the [COLOR2]
        bounding box.",
        "During the entire video, the distance traveled by the object annotated with the
        [COLOR1] bounding box is greater than that of the object annotated with the
        [COLOR2] bounding box.",
        "In the video, the object annotated with the [COLOR1] bounding box has trav-
        eled farther than the object annotated with the [COLOR2] bounding box."

]

traveled_distance_comparison_negative_answers = [
        "The object annotated with the [COLOR1] bounding box has traveled a shorter
        distance than the object annotated with the [COLOR2] bounding box through-
        out the video.",
        "In the video, the object annotated with the [COLOR1] bounding box has
        moved a shorter distance than the one annotated with the [COLOR2] bound-
        ing box.",
        "The object annotated with the [COLOR1] bounding box covers less distance
        throughout the video compared to the object annotated with the [COLOR2]
        bounding box.",
        "During the entire video, the distance traveled by the object annotated with
        the [COLOR1] bounding box is less than that of the object annotated with the
        [COLOR2] bounding box.",
        "In the video, the object annotated with the [COLOR1] bounding box has trav-
        eled a shorter distance than the object annotated with the [COLOR2] bounding
        box."

]
```

Table 33: **Question templates for Traveling Speed Comparison.**

```
traveling_speed_comparison_common_prompt = f"""The video lasts for [SEC-
ONDS] seconds, and [FRAMES] frames are uniformly sampled from it. These frames are located
at [SECOND1]s,[SECOND2]s,[SECOND3]s, ... . Please answer the following questions related to
this video.

There are two objects annotated with [COLOR1] and [COLOR2] bounding boxes in the video. """

traveling_speed_comparison_positive_questions = [
        "Which object moves at a higher speed in the video: the one with the
        [COLOR1] bounding box or the one with the [COLOR2] bounding box?",
        "Which object moves faster throughout the video, the object annotated with the
        [COLOR1] bounding box or the [COLOR2] bounding box?",
        "Which object maintains a greater speed in the video: the one annotated with
        the [COLOR1] bounding box or the [COLOR2] bounding box?",
        "Between the objects annotated with the [COLOR1] and [COLOR2] bounding
        boxes, which one moves at a higher speed throughout the video?",
        "Between the two objects, one annotated with the [COLOR1] bounding box
        and the other with the [COLOR2] bounding box, which one has a higher speed
        during the entire video?",
        "Which object, the one annotated with the [COLOR1] bounding box or the
        [COLOR2] bounding box, has a greater average speed in the video?"
]

traveling_speed_comparison_negative_questions = [
        "Which object moves at a lower speed in the video: the one with the [COLOR1]
        bounding box or the one with the [COLOR2] bounding box?",
        "Which object moves more slowly throughout the video, the object annotated
        with the [COLOR1] bounding box or the [COLOR2] bounding box?",
        "Which object maintains a slower speed in the video: the one annotated with
        the [COLOR1] bounding box or the [COLOR2] bounding box?",
        "Between the objects annotated with the [COLOR1] and [COLOR2] bounding
        boxes, which one moves at a slower speed throughout the video?",
        "Between the two objects, one annotated with the [COLOR1] bounding box
        and the other with the [COLOR2] bounding box, which one has a slower speed
        during the entire video?",
        "Which object, the one annotated with the [COLOR1] bounding box or the
        [COLOR2] bounding box, has a lower average speed in the video?"
]
```

Table 34: **Answer templates for Traveling Speed Comparison.**

```
traveling_speed_comparison_positive_answers = [
        "The object annotated with the [COLOR1] bounding box has moved at a faster
        speed than the object annotated with the [COLOR2] bounding box throughout
        the video.",
        "In the video, the object annotated with the [COLOR1] bounding box moves
        faster than the one annotated with the [COLOR2] bounding box.",
        "The object annotated with the [COLOR1] bounding box maintains a higher
        speed throughout the video compared to the object annotated with the
        [COLOR2] bounding box.",
        "During the entire video, the speed of the object annotated with the [COLOR1]
        bounding box is greater than that of the object annotated with the [COLOR2]
        bounding box.",
        "In the video, the object annotated with the [COLOR1] bounding box has
        moved faster than the object annotated with the [COLOR2] bounding box."
]

traveling_speed_comparison_negative_answers = [
        "The object annotated with the [COLOR1] bounding box has moved at a slower
        speed than the object annotated with the [COLOR2] bounding box throughout
        the video.",
        "In the video, the object annotated with the [COLOR1] bounding box moves
        more slowly than the one annotated with the [COLOR2] bounding box.",
        "The object annotated with the [COLOR1] bounding box maintains a lower
        speed throughout the video compared to the object annotated with the
        [COLOR2] bounding box.",
        "During the entire video, the speed of the object annotated with the [COLOR1]
        bounding box is less than that of the object annotated with the [COLOR2]
        bounding box.",
        "In the video, the object annotated with the [COLOR1] bounding box has
        moved more slowly than the object annotated with the [COLOR2] bounding
        box."
]
```

Table 35: **Question templates for Movement Direction Comparison.**

```
movement_direction_comparison_common_prompt = f"""The video lasts for [SEC-
ONDS] seconds, and [FRAMES] frames are uniformly sampled from it. These frames are located
at [SECOND1]s,[SECOND2]s,[SECOND3]s, ... . Please answer the following questions related to
this video.

There are two objects annotated with [COLOR1] and [COLOR2] bounding boxes in the video. """

movement_direction_comparison_positive_questions = [
        "Is the object annotated with the [COLOR1] bounding box moving in the same
        direction as the object annotated with the [COLOR2] bounding box in the
        video?",
        "Is the object annotated with the [COLOR1] bounding box heading in the same
        direction as the object annotated with the [COLOR2] bounding box throughout
        the video?",
        "Are the object annotated with the [COLOR1] bounding box and the object an-
        notated with the [COLOR2] bounding box moving in the same direction during
        the video?",
        "In the video, does the object with the [COLOR1] bounding box move in the
        same direction as the object with the [COLOR2] bounding box?",
        "Are the objects annotated with the [COLOR1] and [COLOR2] bounding boxes
        traveling in the same direction during the entire video?"
]

movement_direction_comparison_negative_questions = [
        "Is the object annotated with the [COLOR1] bounding box moving in a differ-
        ent direction from the object annotated with the [COLOR2] bounding box in
        the video?",
        "Is the object annotated with the [COLOR1] bounding box heading in a dif-
        ferent direction from the object annotated with the [COLOR2] bounding box
        throughout the video?",
        "Are the object annotated with the [COLOR1] bounding box and the object
        annotated with the [COLOR2] bounding box moving in a different direction
        during the video?",
        "In the video, does the object with the [COLOR1] bounding box move in a
        different direction from the object with the [COLOR2] bounding box?",
        "Are the objects annotated with the [COLOR1] and [COLOR2] bounding boxes
        traveling in different directions during the entire video?"
]
```

Table 36: **Answer templates for Movement Direction Comparison.**

```
movement_direction_comparison_true_positive_answers = [
        "Yes, the object annotated with the [COLOR1] bounding box is moving in
        the same direction as the object annotated with the [COLOR2] bounding box
        throughout the video.",
        "Indeed, the object annotated with the [COLOR1] bounding box is heading in
        the same direction as the object annotated with the [COLOR2] bounding box
        during the entire video.",
        "Correct, the object annotated with the [COLOR1] bounding box and the object
        annotated with the [COLOR2] bounding box are moving in the same direction
        in the video."
]

movement_direction_comparison_true_negative_answers = [
        "No, the object annotated with the [COLOR1] bounding box is moving in
        the same direction as the object annotated with the [COLOR2] bounding box
        throughout the video.",
        "Actually, the object annotated with the [COLOR1] bounding box is heading in
        the same direction as the object annotated with the [COLOR2] bounding box
        during the entire video.",
        "Incorrect, the object annotated with the [COLOR1] bounding box and the ob-
        ject annotated with the [COLOR2] bounding box are moving in the same di-
        rection in the video."
]

movement_direction_comparison_false_positive_answers = [
        "No, the object annotated with the [COLOR1] bounding box is moving in a dif-
        ferent direction from the object annotated with the [COLOR2] bounding box
        throughout the video.",
        "Actually, the object annotated with the [COLOR1] bounding box is heading in
        a different direction from the object annotated with the [COLOR2] bounding
        box during the entire video.",
        "Incorrect, the object annotated with the [COLOR1] bounding box and the ob-
        ject annotated with the [COLOR2] bounding box are moving in a different
        direction in the video."
]

movement_direction_comparison_false_negative_answers = [
        "Yes, the object annotated with the [COLOR1] bounding box is moving in a
        different direction from the object annotated with the [COLOR2] bounding box
        throughout the video.",
        "Indeed, the object annotated with the [COLOR1] bounding box is heading in
        a different direction from the object annotated with the [COLOR2] bounding
        box during the entire video.",
        "Correct, the object annotated with the [COLOR1] bounding box and the object
        annotated with the [COLOR2] bounding box are moving in a different direction
        in the video."
]
```

