# OpenReview forum: "ST-VLM: Kinematic Instruction Tuning for Spatio-Temporal Reasoning in Dynamic Videos"
_ICLR.cc/2026/Conference — Submitted to ICLR 2026_

### Official Review · Reviewer_Lsay · 2025-10-28

**Soundness:** 3
**Presentation:** 3
**Contribution:** 2
**Rating:** 2
**Confidence:** 2

**Summary:**

The paper describes a new vision dataset, that was generated synthetically by augmenting existing datasets with 3D object position and movement information (such as the objects’ movement directions, speeds, etc.). It is based in part on existing datasets that include this type of information in some form and in part on the use of 4D reconstruction methods to video datasets where such information is not included.  The resulting dataset is used to finetune a vision-language model, which performs better at predicting this kind of information than models that were not fine-tuned on this kind of data.

**Strengths:**

The paper introduces a novel dataset. The proposed dataset does what it promises to do. It enables a model finetuned on it to perform comparably well at the task of predicting speeds, directions, etc. of objects shown in the video.

**Weaknesses:**

The dataset focuses on domains like sports and driving, where object position and motion information can be readily extracted. This raises the question what the benefit of distilling this information into an existing vision language model is, instead of relying on existing approaches which would use an external tool that simply extracts such information at inference time. While this question is answered partly in section 6.3 on emergent capabilities, that section is very preliminary and highlights just a few examples qualitatively.

The method seems to rely on relatively clean scenarios that are free from occlusions, camera motions.

The task of predicting speed is inherently ill-posed, suggesting that applying the model to scenarios that are slightly out of the ordinary with respect to object geometry or size, for example showing remote controlled toy cars, would make the model give drastically incorrect results? It would be good to better understand the limitations of this approach considering this.

**Questions:**

Will the dataset and model be publicly released?

How would the performance of the model vary if the framerate of a video is varied? Is robustness to such variations not important, especially when considering this as a foundational skill of a vision-language model?

What about extraordinary scenarios, like videos showing toy cars mentioned above?

(minor) The figures (especially Figure 1 and 2) are good for on-screen viewing but kind of hard to see in a printout.

---

> ### Author Response · Authors · 2025-11-20
> **Response to Reviewer Lsay**
>
> We sincerely thank the reviewer for the constructive review of our paper. We believe that incorporating this constructive feedback significantly enhances the quality of the paper. Below, we provide the response to the given comments in detail:
>
> ---
>
> **Q1. Clarify the motivation for integrating object position and motion understanding into Vision-Language Models (VLMs).**
>
> Directly integrating object position and motion understanding into VLMs offers two key advantages over relying on external tools.
>
> First, enriching the model with objects’ positional and motion cues improves general video understanding, yielding substantial gains in spatial and temporal perception (Tab. 7, 8 and Fig. 6).
> For example, in Figure 10a, when asked "What happens to the object after it is placed on the slanted plane?", ST-VLM accurately captures the object motion and provides the correct answer, whereas the LLaVA-OneVision baseline fails.
> This integrated design also enables emergent multi-step reasoning grounded in kinematic information, as evidenced by ST-VLM correctly answering questions such as "Calculate how long it takes for the object to travel from New York to Los Angeles at the current speed" (Figures 1, 7, and 9).
>
> Second, external 4D reconstruction tools are difficult to integrate into flexible pipelines capable of handling diverse question formats, limiting their practicality, whereas our approach naturally deals with a wide range of question types via instruction tuning.
> Moreover, such tools incur substantial computational overhead, approximately 400 seconds per video, while our integrated approach performs inference in 1.7 seconds, achieving a 235× speedup.
>
> Notably, similar integration strategies have proven effective for spatial reasoning in prior work (SpatialVLM [1], Spatial-RGPT [2]), such as estimating absolute object size and inter-object distances.
> Our approach extends this paradigm to the spatio-temporal domain by incorporating kinematic information.
>
> [1] Boyuan et al., SpatialVLM: Endowing Vision-Language Models with Spatial Reasoning Capabilities., CVPR 2024.
>
> [2] An-Chieh et al., SpatialRGPT: Grounded Spatial Reasoning in Vision Language Models., NeurIPS 2024.
>
> ---
>
> **Q2. Is the model robust in diverse scenarios, e.g., occlusions, camera motions, varying FPS?**
>
> Our model demonstrates robustness across a wide range of challenging scenarios, including object occlusion, multi-object scenarios, dynamic scenes with substantial camera motion, small object sizes, and varying FPS, as shown in Tab. 11–15.
> For example, in Fig. 5, the model successfully answers the question even when the video contains numerous small or partially occluded objects.
> Furthermore, by providing timestamp information as part of the input prompt, the model effectively handles variations in FPS, as evidenced in Tab. 15.
>
> ---
>
> **Q3. Provide the result of the RC car.**
>
> We observe strong generalization even in RC car scenarios, which we attribute to the pseudo-labeled data sourced from diverse domains beyond road scenes.
> We add the RC car result in Fig. 12 of the supplement.
> As illustrated in Fig. 12, when asked "Can you calculate the total distance covered by the red RC car throughout the entire video?", ST-VLM estimates the traveled distance as 12.73 m, which appears to be within a reasonable range for the actual distance.
>
> ---
>
> **Q4. Will all curated datasets and benchmarks be made publicly available to the research community?**
>
> Yes, we will release all datasets, benchmarks, models, and code as open-source upon paper acceptance.
>
> ---
>
> **Q5. Improve the figure quality.**
>
> Thank you for the suggestion. We will improve the figure quality in the final version.

---

> > ### Comment · Reviewer_Lsay · 2025-11-26
> >
> > Thank you for these clarifications. I have a few follow-up questions and comments:
> >
> > - Counter-intuitively, the results in Tab 11-15 show that occlusions, object quantity and camera motions improve accuracy (and very significantly so) rather than reducing it. This seems very strange, given the significant changes, and it seems worrisome in the absence of any further analysis or explanation? Please do let me know in case I am missing something here.
> >
> > - Did you verify that the prompts used for filtering for occlusion, camera motion, etc. truly do what they are supposed to do? For example, I don't see how the following prompt would be accurately determining camera movement: "Camera movement: Do the video frames transition smoothly, without noticeable temporal discontinuities? Respond with ‘Yes’ or ‘No’."
> >
> > - In fact, I am doubtful that most existing vision-language model would be good at detecting camera motion at all. Which LVM was used for the filtering?

---

> ### Author Response · Authors · 2025-11-27
> **Response to Reviewer Lsay**
>
> We sincerely thank constructive questions and comments. Below, we provide the response to the given comments in detail:
>
> ---
>
> **Q6. Discuss the performance under diverse scenarios.**
>
> Occlusion scenarios in STKit-Bench primarily fall under the *multi-object* task category, which is formulated as a binary classification task and generally achieves higher accuracy than the *single-object* category.
> Consequently, performance in occlusion scenarios often appears relatively high in Tab. 11.
> As shown in the table below, 129 occlusion scenes belong to the multi-object category, whereas only 26 belong to the single-object category.
> This imbalance leads to higher total accuracy for occlusion scenarios, even though non-occlusion scenarios exhibit better performance within each individual category.
>
> | occlusion | single object | multiple objects | total |
> | --- | --- | --- | --- |
> | not occluded | 46.9% (363 / 774) | 74.3% (350 / 471) | 57.3% (713 / 1245) |
> | occluded | 26.9% (7 / 26) | 73.6% (95 / 129) | 65.8% (102 / 155) |
>
> Additionally, for camera movement and object quantity in Tab. 12 and 13, it is expected that dynamic camera motion and a larger number of objects lead to performance degradation.
> For object size in Tab. 14, the small- and medium-object scenarios yield better performance than the large-object scenario.
> We conjecture that this is because estimating kinematic quantities, such as traveled distance or moving direction, is easier when sufficient background information is available, whereas large objects often dominate the frame and provide fewer contextual cues.
> We have updated Tab. 11–14 to include detailed statistics.
>
> ---
>
> **Q7. Which VLM is used for model-based filtering?**
>
> We use LLaVA-OneVision, which also serves as our backbone VLM, for filtering.
> To illustrate how the model identifies low-quality samples, we provide examples of filtered cases in Fig. 14 of the supplement.
> In Fig. 14a, the model detects a significant scene transition between the first and second frames.
> In Fig. 14b, it successfully identifies object occlusion, demonstrating that our filtering strategy effectively removes low-quality samples.
> Overall, applying this filtering strategy yields a 5.7% performance improvement.
>
> ---
>
> We sincerely thank you for the constructive feedback, which has significantly strengthened our paper, and we hope these results help address your concerns.
> We would greatly appreciate any further feedback.

---

> > ### Comment · Reviewer_Lsay · 2025-11-27
> >
> > Thank you for the clarifications.
> >
> > This may be minor, but I still find it hard to believe that Llava-OneVision prompted with
> > "Camera movement: Do the video frames transition smoothly, without noticeable temporal discontinuities? Respond with ‘Yes’ or ‘No’."
> > would be able to detect camera motions reliably.

---

> ### Author Response · Authors · 2025-11-28
> **Response to Reviewer Lsay**
>
> **Q8. Discuss the camera motion detection.**
>
> With the camera movement prompt, we observe that the VLM can detect *significant scene transitions*, which are unsuitable for estimating kinematic quantities.
> Additionally, by incorporating this capability into our model-based filtering, together with rule-based and task-specific filtering, we effectively remove low-quality samples, leading to a performance improvement by 5.7\%
>
> ---
>
> We thank you again for the constructive feedback, and we would greatly appreciate further feedback.

---

### Official Review · Reviewer_8N6h · 2025-10-31

**Soundness:** 3
**Presentation:** 3
**Contribution:** 3
**Rating:** 6
**Confidence:** 3

**Summary:**

This paper is doing the task that let the vlm understand kinematic spatio-temporal reasoning, such as an object's speed or traveled distance. The authors introduce ST-VLM, a new model trained via kinematic instruction tuning. To enable this, they create the STKit dataset, which contains video-based questions about speed, distance, and movement direction. They also introduce novel pseudo-labeling pipeline that uses 4D reconstruction to generate this kinematic data from unlabeled videos. On STKit-Bench, ST-VLM substantially outperforms baselines like GPT-5 and demonstrates emergent multi-step reasoning capabilities.

**Strengths:**

- The paper is well writing with clear logic and good motivation.
- The three-stage filtering strategy (rule-based, general model-based, and task-specific model-based) is comprehensive. It demonstrably improves the quality of pseudo-labels.
- The ablation in Table 6 clearly validates the authors' design choices, showing that both the 3D ground-truth data and the filtered pseudo-label data contribute significantly to the final model's performance.
- Experiment and dataset building method are detailed.

**Weaknesses:**

- The pseudo-label is not accurate. While the filtering pipeline is effective, a 29% mean error rate remains in the pseudo-labeled traveled distance data.
- See in questions.

**Questions:**

- How about using data generated in simulator? What is the advantage of simulator data with the 4D model annotated data? In my view is that the 4D model annotated data may not accurate but the video distribution is from the real data, but the simulator data is accurate but is not photo realistic.
- Recently some work also using vlm do depth estimation such as DepthLM[1]. I am wondering what is their method different with this paper(and I think ST-VLM also can do depth estimation). I am wondering if the authors could provide the results of ST-VLM on depth estimation tasks.
- Open source plan?

---

> ### Author Response · Authors · 2025-11-20
> **Response to Reviewer 8N6h**
>
> We sincerely thank the reviewer for the constructive review of our paper. We believe that incorporating this constructive feedback significantly enhances the quality of the paper. Below, we provide the response to the given comments in detail:
>
> ---
>
> **Q1. Discuss the effectiveness of pseudo-labeled data.**
>
> We observe that training with pseudo-labeled data, even with a 29% mean error rate, still leads to notable performance improvements.
> As shown in Tab. 6, incorporating pseudo-labeled data increases accuracy by 7.0% compared to using only GT-labeled data.
> This observation is consistent with prior findings [1], which demonstrate that even noisy data can enhance final performance under long training schedules.
>
> [1] Sachin et al., Scaling Laws for Data Filtering -- Data Curation cannot be Compute Agnostic., CVPR 2024.
>
> ---
>
> **Q2. How effective is simulation data?**
>
> Thank you for the suggestion! We construct kinematic instruction data using two simulation datasets, VKITTI [2] and GTA V [3], to evaluate the effectiveness of simulation-based videos for real-world evaluation scenarios.
> The table below reports the performance of ST-VLM trained with simulation data, showing that simulation alone provides performance gains, although the improvement is smaller compared to training with real-world videos, *i.e.*, pseudo-labeled and GT-labeled data, due to the domain gap.
>
> |  | Acc |
> | --- | --- |
> | LLaVA-OneVision | 26.4 |
> | ST-VLM (simulation) | 29.1 |
> | ST-VLM (pseudo) | 46.1 |
> | ST-VLM (GT) | 52.6 |
>
> [2] Adrien et al., Virtual Worlds as Proxy for Multi-Object Tracking Analysis., CVPR 2016.
>
> [3] Stephan R. et al., Playing for Benchmarks., ICCV 2017.
>
> ---
>
> **Q3. Performance on depth estimation.**
>
> Great suggestion! Surprisingly, our kinematic instruction tuning enables ST-VLM to implicitly acquire depth understanding as part of its kinematic understanding process, even though our dataset does NOT contain any explicit depth estimation samples.
> In contrast, DepthLM is a specialized model dedicated solely to depth estimation and does not generalize to kinematic understanding tasks.
> As shown in the table below, ST-VLM achieves 21.2% accuracy and a 10.2 m MAE on DepthLMBench (depth estimation) despite no explicit depth-specific supervision, whereas DepthLM cannot estimate the object kinematic quantities required in STKit-Bench, underscoring the broader reasoning capability of our model.
>
> |  | STKit-Bench (acc) | DepthLMBench (acc / MAE) |
> | --- | --- | --- |
> | DepthLM | 13.2 | 19.7 / **9.9** |
> | LLaVA-OneVision | 26.4 | 7.5 / 45.9 |
> | ST-VLM | **58.2** | **21.2** / 10.2 |
>
> ---
>
> **Q4. Will all curated datasets and benchmarks be made publicly available to the research community?**
>
> Yes, we will release all datasets, benchmarks, models, and code as open-source upon paper acceptance.

---

### Official Review · Reviewer_AMF5 · 2025-11-01

**Soundness:** 3
**Presentation:** 3
**Contribution:** 3
**Rating:** 6
**Confidence:** 4

**Summary:**

The paper introduces ST-VLM, a vision-language model enhanced for spatio-temporal reasoning through kinematic instruction tuning using the STKit dataset and STKit-Bench benchmark. It aims to address a key limitation of existing VLMs -- poor handling of dynamic object kinematics -- by leveraging datasets with 3D and 4D information. The paper includes extensive evaluation of state of the art models on STKit-Bench and the ST-VLM shows promising signs of improved spatio-temporal understanding.

**Strengths:**

* The proposed STKIT dataset and benchmark is novel and interesting.

* The paper shows that fine-tuning on the proposed STKIT dataset leads to improved performance over the plan LLaVA-OneVision-7B model on PerceptionTest, MVBench, VideoMME, MLVU, NExT-QA. Which shows improved spatio-temporal understanding.

* The paper includes extensive evaluation of state of the art models on STKit-Bench.

**Weaknesses:**

* The proposed STKIT-BENCH includes a vast majority of questions (92.9%) from the autonomous diving domain. This limits the diversity of the benchmark. Due to the heavy reliance on the autonomous driving domain, the paper should include comparison to popular benchmarks such as "DriveLM: Driving with Graph Visual Question Answering, ECCV 2024".

* For 3D understanding of street scenes and answering kinematic questions on distances, it is critical to have access the camera parameters. The proposed model is only able to perform well because the STKIT dataset and benchmark uses the same datasets: Nuscenes and Argoverse. Training on these datasets allows the model to memorize the camera parameters and thus perform well on the evaluation data. The STKIT-BENCH should include datasets such as Cityscapes for true zero-shot evaluation.

* The evaluation in Table 7 and Table 8 should include state of the art approaches such as VideoLLAMA3 and Qwen-2.5-VL.

* The paper should discuss prior work on grounding to fine-grained spatio-temporal visual information in videos such as: "Look, Remember and Reason: Grounded reasoning in videos with language models, ICLR 2024"; "Fine-grained Spatiotemporal Grounding on Egocentric Videos, ICCV 2025".

**Questions:**

* The choice of data sources for STKIT-BENCH should be motivated in more detail.
* The effect of the overlap between STKIT dataset and benchmark should be explain in more detail.
* Prior work in grounding to spatio-temporal visual information in videos should be explain in more detail.

---

> ### Author Response · Authors · 2025-11-20
> **Response to Reviewer AMF5**
>
> We sincerely thank the reviewer for the constructive review of our paper. We believe that incorporating this constructive feedback significantly enhances the quality of the paper. Below, we provide the response to the given comments in detail:
>
> ---
>
> **Q1. The motivation of the source data choice.**
>
> Our STKit-Bench primarily evaluates kinematic quantities, which constitute fundamental capabilities for general spatio-temporal understanding in dynamic videos.
> Therefore, we primarily target domains in which kinematic quantities are the primary subject of interest, such as driving scenarios and sports environments.
> In contrast, most existing autonomous driving benchmarks [1, 2] focus on driving-specific scenarios, such as planning and decision-making, rather than core kinematic reasoning.
>
> In our experiments, leveraging our kinematic instruction tuning dataset, ST-VLM demonstrates improved general video understanding, particularly in spatial and temporal perception, as shown in Tab. 7, 8, and Fig. 6.
> For example, in Fig. 10a, when asked "What happens to the object after it is placed on the slanted plane?", ST-VLM accurately captures the object motion and correctly answers, whereas the LLaVA-OneVision baseline fails to provide the correct response.
>
> [1] Yunsong et al., Embodied understanding of driving scenarios., ECCV 2024.
>
> [2] Chonghao et al., DriveLM: Driving with Graph Visual Question Answering., ECCV 2024.
>
> ---
>
> **Q2. Performance on out-of-domain settings.**
>
> Good Point! We have already evaluated the out-of-domain generalization within STKit-Bench by incorporating NuPlan data, which is not included in the training set, constituting 74.8% of the evaluation benchmark and featuring distinct camera intrinsics and extrinsics, road scenes, weather conditions, illumination, and locations.
>
> We further conduct an additional evaluation on the Waymo dataset [3], which also employs different camera parameters at different road scenes.
> The table below shows detailed results across in-domain and out-of-domain settings, demonstrating the robustness of our model to variations in camera configurations and road scenes across datasets.
>
> |  | in-domain (NuScenes & Argoverse) | out-of-domain (NuPlan) | out-of-domain (Waymo) |
> | --- | --- | --- | --- |
> | LLaVA-OneVision | 35.4 | 26.0 | 30.8 |
> | ST-VLM | 63.4 (+28.0) | 55.6 (+29.6) | 58.4 (+27.6) |
>
> We will include Waymo data into STKit-Bench in the final version to enable a more comprehensive and robust evaluation.
>
> [3] Pei et al., Scalability in Perception for Autonomous Driving: Waymo Open Dataset., CVPR 2020.
>
> ---
>
> **Q3. Compare with prior works.**
>
> Our work introduces instruction data grounded in fundamental kinematic quantities, such as object speed and traveled distance, derived from 4D information, *i.e.*, 3D object trajectories across frames, while [4, 5] aim to enhance spatio-temporal understanding in videos using only high-level 2D information.
> Specifically, [4] introduces a novel pixel-level, fine-grained spatio-temporal grounding benchmark in egocentric videos, while [5] proposes an elegant three-step video reasoning framework (Look, Remember, Reason) incorporating a two-stream video encoder and spatio-temporal attention.
>
> [4] Shuo et al., Fine-grained Spatiotemporal Grounding on Egocentric Videos., ICCV 2025.
>
> [5] Apratim et al., Look, Remember and Reason: Grounded reasoning in videos with language models., ICLR 2024.
>
> ---
>
> **Q4. Include state-of-the-art baseline methods.**
>
> Thank you for the suggestion. Since our backbone model is LLaVA-OneVision, ST-VLM achieves a 3.0% improvement in average accuracy over LLaVA-OneVision, and we expect similar gains when applied to stronger backbone models.
> Due to limited training resources for larger state-of-the-art models, we will include results from fine-tuned versions of these models in the final version.

---

### Official Review · Reviewer_hL4E · 2025-11-02

**Soundness:** 2
**Presentation:** 3
**Contribution:** 2
**Rating:** 4
**Confidence:** 4

**Summary:**

The paper aims to strengthen the spatio-temporal reasoning capability of current VLMs. To this end, the authors introduce STKit, a large-scale dataset with seven types of motion-related question–answer pairs generated through a 4D reconstruction and pseudo-labeling pipeline, and STKit-Bench, a benchmark for quantitative evaluation. By tuning LLaVA-OneVision with these instructions, it achieves performance improvement on STKit-Bench and video understanding benchmarks.

**Strengths:**

1.	The paper is well-written.
2.	I appreciate the authors’ efforts to curate data for spatio-temporal reasoning.
3.	The results are promising after finetuning the baseline with the curated data.

**Weaknesses:**

1.	For STKit-Bench, how to ensure annotation quality, especially annotation related to the distance annotation? Meanwhile, how to ensure its diversity to cover the real-world scenes?
2.	Both training data and testing data come from the same data annotation pipeline. How could the authors deal with domain overlap?
3.	The method does not explicitly model physical kinematics. Therefore, “kinematic instruction tuning” may not be a good description.
4.	Could the authors show some evidence that the models learns the spatio-temporal reasoning rather than static correlations?
5. Will all curated datasets/benchmarks be public to the community?

**Questions:**

See the questions in Weakness part.

---

> ### Author Response · Authors · 2025-11-20
> **Response to Reviewer hL4E**
>
> We sincerely thank the reviewer for the constructive review of our paper. We believe that incorporating this constructive feedback significantly enhances the quality of the paper. Below, we provide the response to the given comments in detail:
>
> ---
>
> **Q1. Discuss the diversity and quality of annotations.**
>
> For GT-labeled videos, we leverage real-world data annotated with LiDAR/SLAM, ensuring both high-quality and high-diversity annotations.
> To further expand diversity, we introduce a pseudo-labeling pipeline applied to additional real-world videos, which increases the number of high-speed cases (object speeds exceeding 60km/h) from 36 to 1,563, a 43× improvement in coverage.
> To preserve annotation reliability while broadening coverage, we apply a three-stage filtering strategy that reduces the mean distance estimation error from 207% to 29%, demonstrating a substantial enhancement in annotation quality.
>
> ---
>
> **Q2. Performance on out-of-domain settings.**
>
> We have already evaluated the out-of-domain generalization within STKit-Bench by incorporating NuPlan data, which is not included in the training set, constituting 74.8% of the evaluation benchmark and featuring distinct camera intrinsics/extrinsics, road scenes, weather conditions, illumination, and locations.
>
> We further conduct an additional evaluation on the Waymo dataset [1], which also employs different camera parameters at different road scenes.
> The table below shows detailed results across in-domain and out-of-domain settings, demonstrating the robustness of our model to variations in camera configurations and road scenes across datasets.
>
> |  | in-domain (NuScenes & Argoverse) | out-of-domain (NuPlan) | out-of-domain (Waymo) |
> | --- | --- | --- | --- |
> | LLaVA-OneVision | 35.4 | 26.0 | 30.8 |
> | ST-VLM | 63.4 (+28.0) | 55.6 (+29.6) | 58.4 (+27.6) |
>
> We will include Waymo data into STKit-Bench in the final version to enable a more comprehensive and robust evaluation.
>
> [1] Pei et al., Scalability in Perception for Autonomous Driving: Waymo Open Dataset., CVPR 2020.
>
> ---
>
> **Q3. Does the model learn genuine spatio-temporal reasoning rather than static correlations?**
>
> Yes, ST-VLM demonstrates improved spatio-temporal reasoning, including complex multi-step reasoning grounded in kinematic quantities, rather than relying on static correlations or dataset memorization.
> For example, in Fig. 1b, given the question "Calculate how long it takes for the object to travel from New York to Los Angeles at the current speed", ST-VLM successfully performs multi-step reasoning that requires integrating (1) commonsense knowledge (distance between cities), (2) kinematic estimation (object speed), (3) relevant equation recall (time = distance / speed), and (4) arithmetic computation, even without explicit training for such complex reasoning.
> Additionally, as shown in Q2, the model exhibits robustness across diverse road scenes (*e.g.*, weather, illumination, and location) and varying camera parameters, indicating genuine spatio-temporal reasoning capabilities.
>
> ---
>
> **Q4. Clarify the meaning of "kinematic instruction tuning".**
>
> *Kinematic instructions* refer to questions and answers that target kinematic quantities (*e.g.*, object speed, traveled distance), which the model must answer based on kinematic cues present in the video.
> Accordingly, *kinematic instruction tuning* refers to the fine-tuning process using these instructions, which distills kinematic knowledge into the model.
>
> ---
>
> **Q5. Will all curated datasets and benchmarks be made publicly available to the research community?**
>
> Yes, we will release all datasets, benchmarks, models, and code as open-source upon paper acceptance.

---

### Author Response · Authors · 2025-11-25
**Global response by authors**

We sincerely appreciate the reviewers for their time and constructive feedback. We have addressed all questions and concerns raised by the reviewers and uploaded a revised version of the manuscript, with changes highlighted in blue for clarity. The major updates are summarized below:

1. Out-of-domain evaluation
* We updated Sec. C.3 to provide quantitative results on out-of-domain settings using the NuPlan and Waymo datasets (Reviewers hL4E and AMF5).

2. Simulation data results
* We updated Sec. C.4 to present quantitative results obtained from simulation data (Reviewer 8N6h).

3. Depth estimation results
* We updated Sec. C.5 to include quantitative depth estimation results and comparisons with DepthLM (Reviewer 8N6h).

4. Extraordinary scenario: RC car
* We updated Sec. D.6 to include a qualitative result illustrating the estimation of the RC car’s traveled distance (Reviewer Lsay).

5. Comparison with prior work
* We updated Sec. 2 by adding comparisons with relevant prior work (Reviewer AMF5).

We sincerely thank all reviewers for their time and thoughtful feedback. We believe our responses have addressed the raised suggestions and questions, and we would be happy to clarify anything further if needed. We greatly appreciate the opportunity for this discussion and look forward to your continued feedback.

---

### Author Response · Authors · 2025-12-02
**Rebuttal Summary for Area Chair**

**To Area Chair,**

We greatly appreciate your efforts, especially under the unusual situation this year, and we are sincerely grateful for your time and careful evaluation. To make your decision easier and provide a clear overview, we summarize our rebuttal as follows:
1. A concise summary of the paper’s core motivation and contributions.
2. A brief summary of each reviewer’s rating, overall attitude, and the concerns we addressed.

---

**1. Motivation and contributions.**

(1) We propose **STKit/STKit-Bench**, a dataset designed to encourage Vision-Language Models (VLMs) to estimate **kinematic quantities** (*e.g.*, object travel distance, moving direction).

(2) To construct STKit, we employ a **LiDAR/SLAM**-based annotation pipeline to obtain high-quality GT kinematic labels, and we introduce a complementary **pseudo-labeling pipeline, followed by a three-stage filtering strategy**, to alleviate the data bottleneck for GT-labeled samples.

(3) By training the backbone VLM, LLaVA-OneVision, on STKit, we develop **ST-VLM**, which demonstrates **strong generalization** across diverse domains and tasks, as well as **emergent capabilities** in complex kinematics-grounded reasoning.

---

**2. Reviewer’s rating, overall attitude, and the concerns we addressed**

We are fortunate to have four engaged reviewers whose comments have substantially strengthened our work.

All reviewers unanimously recognize that our proposed STKit dataset is novel and highly valuable, although some raised concerns regarding domain generalization due to potential overfitting to camera parameters.

With the additional clarifications and experiments conducted during the rebuttal, we believe these concerns have been thoroughly addressed, and we kindly invite the Area Chair to verify this.

1. Reviewer **hL4E** (rating 4, confidence 4)
    - Attitude / Key focus
        - Acknowledges the **effectiveness of the proposed dataset**.
        - Requests clarification regarding the **annotation quality and diversity**.
        - Raises a concern regarding **domain generalization**.
    - Concerns addressed
        - We clarify that the annotations are derived from **LiDAR/SLAM**, which is a **practical GT**, and we **enhance the diversity** through our novel pseudo-labeling pipeline.
        - Through **out-of-domain evaluations** with different camera parameters on NuPlan and Waymo, performance improves by **29.6%** and **27.6%**, respectively.
        - We believe that this **robustness across diverse domains** addresses the reviewer’s concern about **domain generalization**.
2. Reviewer **AMF5** (rating 6, confidence 4)
    - Attitude / Key focus
        - **Clearly supportive** for our dataset and benchmark.
        - Requests comparison with prior works.
        - Concerns in **overfitting to camera parameters**.
    - Concerns addressed
        - We added the comparison with prior works in the revision.
        - Through **out-of-domain evaluations** with different camera parameters on NuPlan and Waymo, performance improves by **29.6%** and **27.6%**, respectively.
        - We believe that this **robustness across diverse domains** addresses the reviewer’s concern about overfitting.
3. Reviewer **8N6h** (rating 6, confidence 3)
    - Attitude / Key focus
        - **Highly positive**.
        - Acknowledges our novel dataset and pseudo-labeling pipeline.
        - Shows interest in the **simulation data**.
        - Asks the **depth estimation** performance.
    - Concerns addressed
        - We provide experiments using **simulation data**, achieving a 2.7\% improvement.
        - We evaluate our ST-VLM on **DepthLMBench**, where it **outperforms DepthLM by 1.5\%**, despite not being explicitly trained for depth estimation.
        - We believe that these results substantively reinforce the reviewer’s positive assessment of our approach.
4. Reviwer **Lsay** (rating 2, confidence 2)
    - Attitude / Key focus
        - Concerns in **motivation.**
        - Actively engages discussion.
        - Asks the performance under **diverse scenarios**.
        - These **concerns were addressed** during the discussion.
    - Concerns addressed
        - We **clarify the motivation** for integrating object position and motion understanding into VLMs.
        - We provide additional results on kinematic estimation in **extraordinary scenarios**, such as RC car settings.
        - We report detailed experiments under **occlusions**, **dynamic camera motion**, **varying object quantities**, or **different FPS**, to validate the **robustness** of our model.
        - The reviewer **acknowledges our clarifications**, and we believe these results sufficiently address their concerns, leading to **strengthened support**.

---

We hope this concise summary helps you decision process. We are, of course, happy to be judged strictly on these clarified settings and results, and we sincerely thank you again for your time and careful consideration.

---

### Meta-Review · Area_Chair_5Lkq · 2026-01-06

**Summary:**

The paper introduced a new dataset/benchmark for VLMs to improve/evaluate their spatial-temporal reasoning capabilities. This is a timely topic, and the tasks are definitely of interest to the community, which all reviewers agree. The reviewers are mainly concerned about the diversity of the data (since it largely focused on self-driving scenarios), the baseline models the authors compared to, and the potential generalization capability. During the discussion period, the authors argued that this focus is intentional, since kinematic quantities are the primary objects of interest for autonomous vehicles. The authors also provided generalization experiments, yet these remain largely confined to self-driving scenes. Overall, the authors did not fully address the reviewers’ concerns. At the end, the reviewers are on the fence. Importantly, no reviewer is willing to champion the paper. After extensive discussion, the AC find the cons outweigh pros and decide to reject the paper. The authors are encouraged to incorporate the feedbacks from the reviewers and submit to a future venue.

**Reviewer Concerns:**

The authors provided clarifications on the annotation pipeline, and argued for their design. They also provide a couple new experiments on generalization across different self-driving datasets.

**Reviewer Scores:**

As mentioned above, the authors' responses partially addressed some concerns from reviewers. However, some fundamental limitations of the paper still remains, eg, the scope, the baselines, etc. I expect the negative reviewers to slightly raise their score, but still remain borderline reject.

---

### Decision · Program_Chairs · 2026-01-26

Reject